# Whole-body replacement of larval myofibers generates permanent adult myofibers in zebrafish

Uday Kumar [1,2], Chun-Yi Fang [1], Hsiao-Yuh Roan [1], Shao-Chun Hsu [1], Chung-Han Wang [1] & Chen-Hui Chen [1✉]

## Abstract

**Drastic increases in myofiber number and size are essential to support vertebrate post-embryonic growth. However, the collective cellular behaviors that enable these increases have remained elusive. Here, we created the *palmuscle* myofiber tagging and tracking system for in toto monitoring of the growth and fates of ~5000 fast myofibers in developing zebrafish larvae. Through live tracking of individual myofibers within the same individuals over extended periods, we found that many larval myofibers readily dissolved during development, enabling the on-site addition of new and more myofibers. Remarkably, whole-body surveillance of multicolor-barcoded myofibers further unveiled a gradual yet extensive elimination of larval myofiber populations, resulting in near-total replacement by late juvenile stages. The subsequently emerging adult myofibers are not only long-lasting, but also morphologically and functionally distinct from the larval populations. Furthermore, we determined that the elimination-replacement process is dependent on and driven by the autophagy pathway. Altogether, we propose that the whole-body replacement of larval myofibers is an inherent yet previously unnoticed process driving organismic muscle growth during vertebrate post-embryonic development.**

**Keywords** Zebrafish; Live Imaging; Fast Myofiber; Post-embryonic Growth
**Subject Category** Development

## Introduction

Post-embryonic growth of vertebrate animals is predominantly driven by the expansion of muscle tissues. At the cellular level, muscle stem cells (or satellite cells) can self-renew or undergo asymmetric division to generate functional myofibers during growth and regeneration (Berberoglu et al, 2017; Gurevich et al, 2016; Nguyen et al, 2017). In addition, muscle stem cells can generate myoblasts, which fuse with existing myofibers to generate the multi-nucleated myofibers that are key to muscle morphogenesis and function (Abmayr and Pavlath, 2012; Chal and Pourquie, 2017; Hromowyk et al, 2020). Despite much progress in understanding how muscle stem cells behave in vitro and in vivo, little is known about the behaviors of individual myofibers that make up the majority of muscle mass. In particular, it remains unknown how thousands of myofibers across different anatomical regions are able to grow in concert to enable the rapid expansion of muscle mass during post-embryonic stages. Progress on this fascinating puzzle has been impeded by technical challenges associated with cell tagging tools and imaging platforms. Up to now, long-term monitoring of each and every myofiber of an actively growing individual has not been achieved in any vertebrate animal model.

Here, we addressed these challenges by creating a panel of myofiber tagging and barcoding tools, and an in toto live imaging platform to track the full developmental trajectories of the entire zebrafish skeletal/fast myofiber population from larval to adult stages. We determined that myofiber growth at the larval stages was predominantly driven by hyperplasia and hypertrophy. Yet, as animals grow into juveniles, we spotted many peculiar, deformed myofibers in otherwise healthy-looking individuals. Through continuous tracking of each deformed and ordinary larval myofiber over its lifetime, we determined that most if not all of the larval myofibers are eventually eliminated and replaced by adult counterparts, each possessing distinct morphologies, functions, and lifespans. Mechanistically, we found that cell-autonomous activation of autophagic cell death is both necessary and sufficient for the elimination and the subsequent new myofiber birth at the voided sites. Thus, we propose that zebrafish successively grow two complete sets of fast myofiber populations, i.e., the larval myofibers that serve as a temporary set and the permanent set of adult myofibers. Taken together, our findings not only provide in toto imaging tools and strategies for unraveling intricate tissue dynamics that might persist in vivo for weeks and months, but they also highlight a whole-body transformation process that may be central to vertebrate muscle development and adult transitions.

## Results

### *palmuscle-Dual* enables in toto monitoring of muscle growth

In zebrafish and other vertebrates, skeletal myofiber growth involves both hyperplasia and hypertrophy. Hyperplasia refers to

[1]Institute of Cellular and Organismic Biology, Academia Sinica, Taipei 11529, Taiwan. [2]Molecular and Biological Agricultural Sciences, Taiwan International Graduate Program, Graduate Institute of Biotechnology, National Chung-Hsing University, Taichung, Taiwan. ✉E-mail: chcchen@gate.sinica.edu.tw

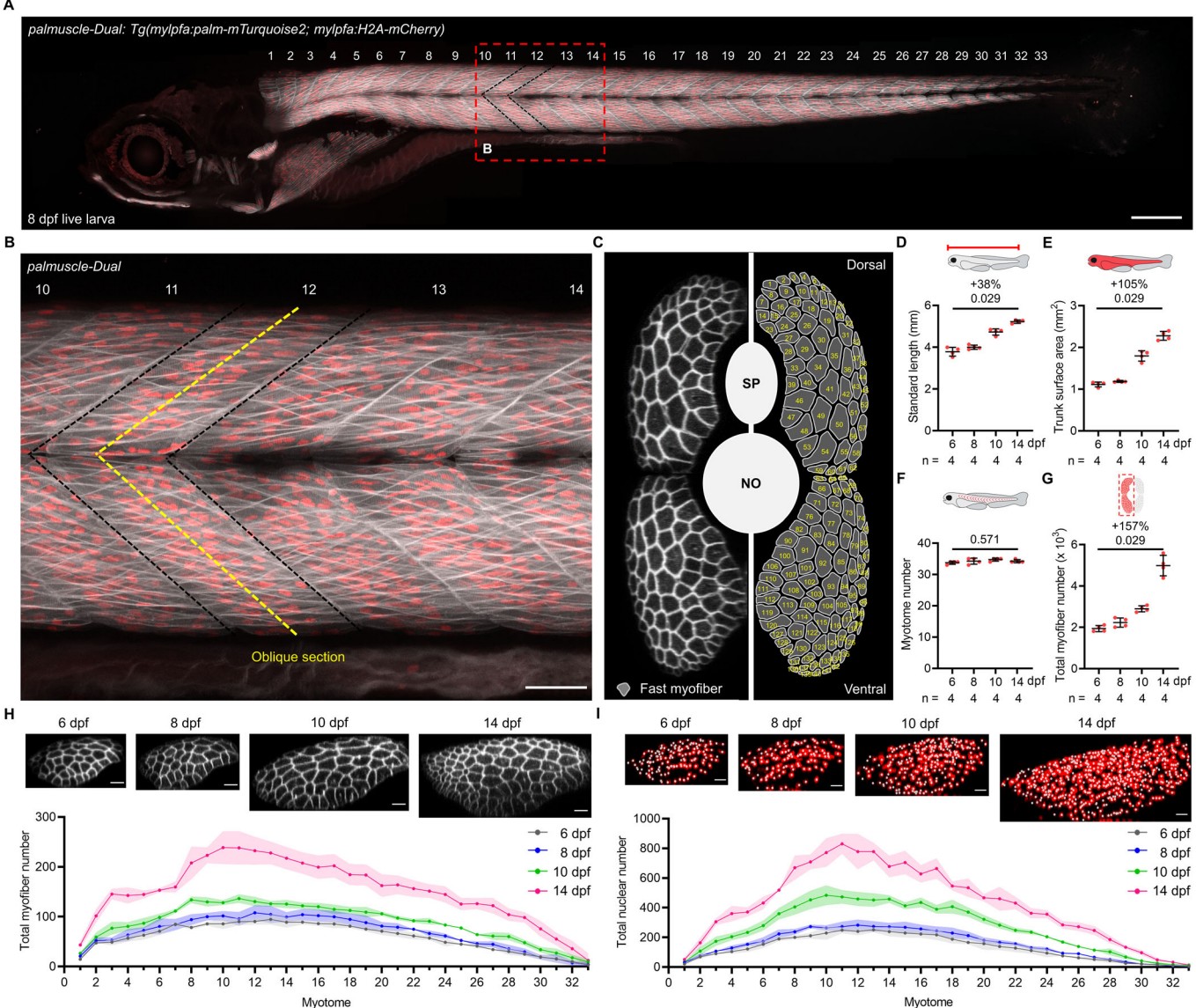

**Figure 1. In toto monitoring of muscle growth in developing zebrafish.**

(**A**) Whole-animal view of a live *palmuscle-Dual* zebrafish larva at 8 dpf (days post-fertilization). Every myotome is marked by a number. Black dashed lines highlight Myotome #12. (**B**) Magnified view of the body region indicated in (**A**) by a red dashed box. Yellow dashed lines mark the planes of optical sectioning used to create the displayed cross-sectional view. (**C**) Representative cross-sectional view of Myotome #12 (left). The schematic drawing on the right mirrors and marks every myofiber within the myotome. SP, spinal cord. NO, notochord. (**D–G**) Larval growth as determined by standard length (**D**), trunk surface area (**E**), myotome number (**F**), and total myofiber number (**G**). (**H, I**) Total myofiber number and the total nuclear number in each myotome from live *palmuscle-Dual* at 6, 8, 10, and 14 dpf. Solid lines indicate mean and colored shadows highlight standard deviation. Representative ventral myotome images from each time point are shown above the graphs. Data from biological replicates are shown as mean ± standard deviation (**D–G**). Significance was examined by two-tailed Mann–Whitney test. Percent differences and *P* values are shown above the horizontal lines for intergroup comparisons. *n* = number of animals (**D–G**). Stitched image (**A**). Scale bars, 200 μm (**A**); 50 μm (**B**); 25 μm (**H, I**). dpf, days post-fertilization. Source data are available online for this figure.

an increase in the number of myofibers, whereas hypertrophy is the volume expansion of individual myofibers. Of note, while myofiber hypertrophy can occur without the addition of nuclei, it is often associated with an increased number of nuclei per myofiber due to fusion of myoblasts with the myofiber (Cramer et al, 2020; Egner et al, 2016; Goh and Millay, 2017). To enable quantitative high-resolution imaging of these two distinct modes of myofiber growth, we created two stable transgenic lines, *Tg(mylpfa:palm-*

*mTurquoise2)* and *Tg(mylpfa:H2A-mCherry)*. With these lines, we expected to visualize dynamic changes in cell number, cell volume, and nuclear number within the entire population of fast myofibers (the predominant skeletal myofiber type in zebrafish). Live imaging and histological examinations provided evidence that the fluorescent tagging of fast myofiber populations in double transgenic animals was comprehensive and exclusive for both transgenes. As such, most if not all fast myofibers in each myotome compartment

along the anterior-posterior (AP) and dorsal-ventral (DV) axes could be captured and counted, but the slow myofiber population was not labeled (Figs. 1A–C and EV1A–D and Movies EV1, 2). To understand how the fast myofiber population behaves during post-embryonic growth, we performed in toto, head-to-tail, live imaging on individual developing zebrafish larvae at 6, 8, 10, and 14 days post-fertilization (dpf). During this 8-day period, the animals exhibited drastic increases in standard length and trunk surface area (38% and 105% by 14 dpf, respectively; Fig. 1D,E). In line with research on early zebrafish development (Morin-Kensicki et al, 2002; Morrow et al, 2017; Patterson et al, 2008), the total myotome number during the post-embryonic growth phase remained largely constant (34 per animal on average; $n = 4$; Fig. 1F); the total myofiber number had a sharp (157%) increase by 14 dpf (from $1942 \pm 148$ to $4988 \pm 497$ per animal; $n = 4$; Fig. 1G). In addition, the extent of myofiber increase within each myotome appeared to be spatially coordinated along the AP axis. The increases were most profound in the middle trunk region of the animal (myotome #10 to #12) at 14 dpf, according to the whole-body measurements of myofiber number and total nuclear number per myotome (Fig. 1H,I). Notably, the increased myofiber numbers from 10 to 14 dpf were tightly correlated with the concurrent increases in nuclear numbers ($R^2 = 0.95$ and 0.96; Fig. EV1E,F). As our tagging strategy, imaging set-up and analysis pipeline allowed for real-time monitoring of two distinct cellular components of a myofiber on a whole-animal scale, we refer to this in toto reporter system as *palmuscle-Dual*.

## Hyperplasia and hypertrophy are coordinated to support rapid muscle growth

To determine how a population of myofibers may respond to dynamic demands for muscle growth under different physiological conditions, we regulated zebrafish larval growth rates by adjusting the rearing conditions (Chan et al, 2022). We used two conditions—slow growth (SG; 1 fish/20 ml) and fast growth (FG; 1 fish/200 ml)—to produce larvae with markedly different body sizes over the course of two days; standard length and trunk surface area at 10 dpf were, respectively, increased by 8% and 23% in FG compared to SG conditions (Fig. 2A–D). Of note, we determined that the acute changes in standard length induced by FG are closely correlated with the progression of developmental stages, in accordance with the normal table of post-embryonic zebrafish development (Parichy et al, 2009) (Fig. EV1G–M). The size differences between the SG and FG conditions persisted at 14 dpf (9% and 31%, respectively; Fig. 2C,D). While there were no changes in total myotome number, the myofiber number was sharply increased in animals kept under the FG condition, indicative of a hyperplasia-mediated muscle growth (32% increase in FG over SG at 14 dpf; Fig. 2E,F). Despite the ongoing increase in the overall myotome volume, the growth of each myofiber appeared to reach a temporal constraint at 14 dpf regardless of the rearing conditions (Fig. 2G–I) and anatomical locations (Fig. EV1N–R). Consistent with this finding, we detected no further increase in the percentages of hyper-nucleated myofibers under the FG condition at 14 dpf (see the "6-to-10" and the "more than 10" groups in Fig. 2J,K). Thus, we concluded that muscle growth in developing zebrafish larvae involves a biphasic mechanism. Our data suggest that when facing a high demand for body growth, both hyperplasia and hypertrophy are initially engaged to

generate more and larger myofibers. However, hypertrophic growth soon reaches a limit, and hyperplasia becomes the dominant mode of muscle growth at later stages.

To track the nuclear number increase during hypertrophy-mediated muscle growth, we applied 3D rendering to optically dissect each myofiber in *palmuscle-Dual* (Movie EV3). Consistent with studies in both mice and zebrafish that showed nuclear numbers may themselves promote hypertrophic growth in myofibers (Cramer et al, 2020; Hromowyk et al, 2020), we found that the nuclear number within a myofiber exhibited a robust, linear correlation with the myofiber volume, regardless of the growth conditions or developmental stage ($R^2 = 0.65$ to 0.97; Fig. EV1S–V). However, we also noted that each additional nucleus is correlated with a larger unit volume increase in the FG condition compared to the SG condition (FG vs. SG: $1.9 \times 10^3$ μm³/nucleus vs. $1.0 \times 10^3$ μm³/nucleus at 10 dpf; $3.2 \times 10^3$ μm³/nucleus vs. $1.5 \times 10^3$ μm³/nucleus at 14 dpf; Fig. EV1T,V). This suggests that growth conditions can regulate the nuclear domain size. Altogether, we conclude that although hypertrophic growth exhibits a growth-rate-independent maximum size limit, the impact of nuclear number augmentation on volume is evidently more pronounced when individuals face higher demands for muscle tissue expansion within a short time period.

## Myofiber tracking reveals *quid pro quo* myofiber birth

When monitoring the population growth of myofibers in intact live animals, we often spotted peculiar myofibers that appeared to be deformed (Fig. 3A,B and Movie EV4). Puzzled by their frequent appearance and occasionally high numbers in otherwise healthy-looking individuals, we conducted time-lapse imaging on the same deformed myofibers in a developing zebrafish larva. Intriguingly, by employing the LifeAct-mScarlet reporter (Bindels et al, 2017), we found that the deformed myofibers initially exhibited distinct signs of detachment between the membrane and cytoskeleton before ultimately undergoing complete dissolution (Fig. 3C,D). Through continuous monitoring of the entire dissolution process at a 12-h interval, we found that the dissolutions led to either no cell replacement or a variable number of in situ myofiber births (Fig. 3E–K and Movie EV5). Among a total of 98 captured events, we determined that the vacant spaces were filled by neighboring myofibers and subsequent creation of new myofiber-myofiber contacts only in a few cases (11%). In sharp contrast, the majority of dissolution events (89%) were associated with addition of one or more new myofibers (Fig. 3L). The removal of a single deformed myofiber in three-dimensional space was spatiotemporally coupled with the birth of either one, two, three, or four myofibers (29%, 21%, 18%, and 13% of dissolution events, respectively; Fig. 3M and EV2A–C). Notably, these dissolution events appeared to exhibit a spatial bias favoring the medial edge compartments (53% in the dorsal myotome and 68% in the ventral myotome; comprising 47 and 51 events, respectively; Fig. EV2D) and were tightly associated with the interstitial *pax7b*-positive muscle stem cells (Pipalia et al, 2016) (90%; $n = 74$ cells; Fig. EV2E–I and Movie EV6). Considering that this removal-replacement process could serve as a rapid yet highly efficient strategy for creating space for additional myofibers, we speculated that "*quid pro quo* myofiber birth" may function as a prevailing mechanism supporting extreme muscle growth during the post-embryonic period.

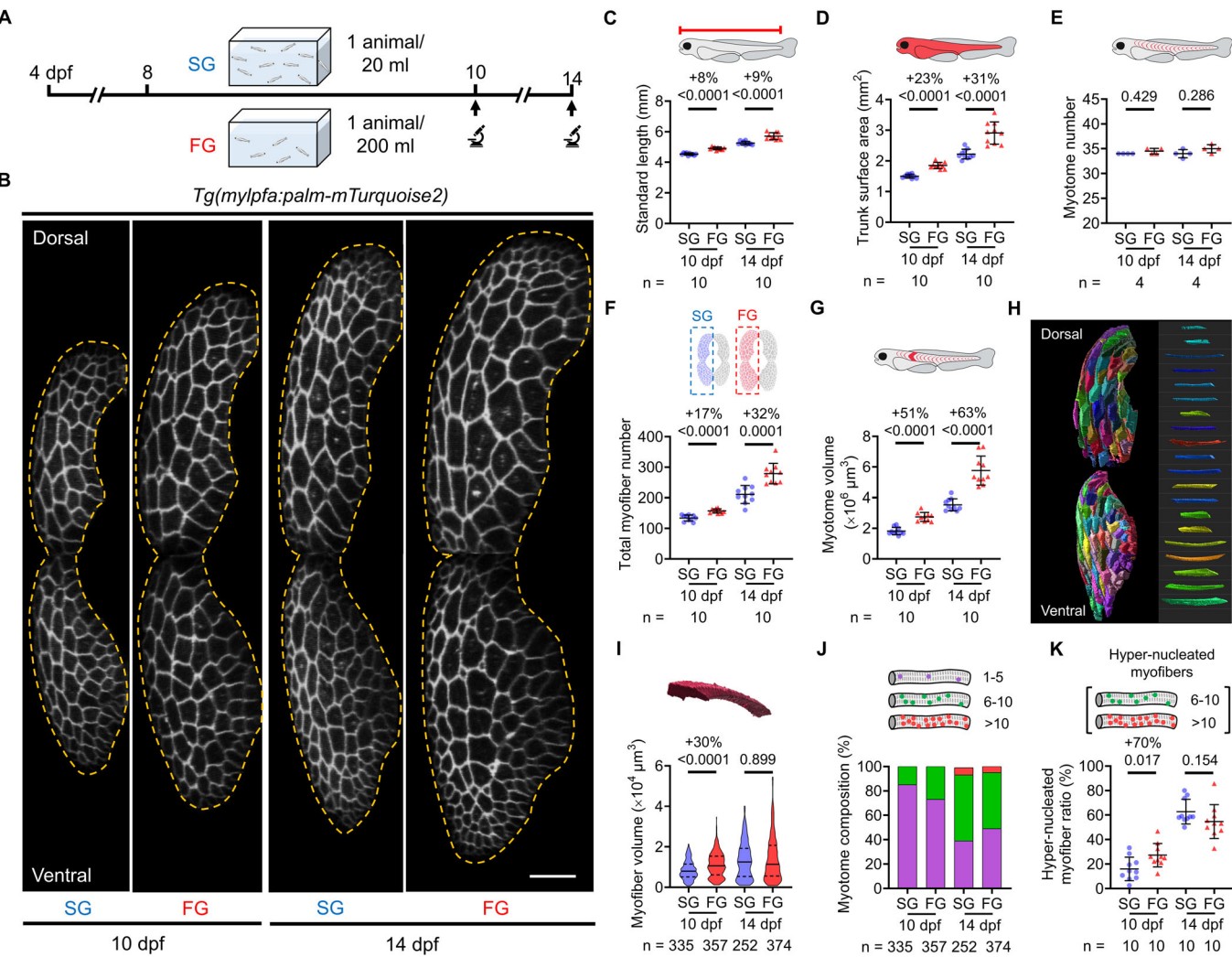

**Figure 2. Drastic increase in myofiber number and volume enables rapid organismic growth.**

(A) Timeline of larval growth manipulation and tracking scheme. SG, slow growth. FG, fast growth. (B) Representative cross-sectional view of Myotome #12 under either the SG or the FG condition at 10 and 14 dpf. Yellow dashed lines mark the boundary of a myotome. (C–G) Larval growth under either the SG and the FG condition as determined by standard length (C), trunk surface area (D), myotome number (E), total myofiber number (F), and myotome volume (G). (H) 3D surface rendering of all myofibers in Myotome #12. Pseudo-colored images highlighting myofiber volume were generated by Imaris. (I–K) Quantitative changes in myofiber volume (I), myotome composition (J), and percentage of the hyper-nucleated myofibers (K). Data from biological replicates are shown as mean ± standard deviation (C–G, K) or violin plots (solid lines, median; dashed lines, quartiles; (I). Significance was examined either by two-tailed Student's t-test (C, D, F, G, K) or two-tailed Mann–Whitney test (E, I). Percent differences and P values are shown above the horizontal lines for intergroup comparisons. $n$ = number of animals (C–G, K) or myofibers (I, J). Scale bar, 50 μm (B). dpf, days post-fertilization. Source data are available online for this figure.

## *palmuscle-Multi* enables multicolor barcoding of larval myofibers

Despite the existence of reports on skeletal muscle death events during early vertebrate development (de Torres et al, 2002; McClearn et al, 1995; Sandri and Carraro, 1999; Webb, 1972), our finding that many mature myofibers are eliminated at larval stages still caught us by surprise. As myofibers are the functional units of muscle tissue, it is generally assumed that the majority of post-embryonic myofibers are stable entities that can live long and prosper through hypertrophy (Hughes et al, 2022). To enable unambiguous, long-term fate mapping of most larval myofibers in an individual, we created a second in toto reporter system

based on the Brainbow technology (Livet et al, 2007; Loulier et al, 2014) (Fig. 4A,B). A notable design feature in this reporter is the use of a membrane-tagging strategy that allows for better visualization of potential dynamic morphological changes in each myofiber (Fig. 4A). Through an extensive screen, we identified one specific floxed line, *palmuscle-Multi*, that suitably illuminated most if not all of the larval myofiber populations across various anatomical regions, such as the pectoral fin, craniofacial, and trunk myofibers (Fig. 4C–E and Movie EV7). Notably, the hues in *palmuscle-Multi* are diverse and well-balanced across RGB color space. In combination with an inducible Cre driver line [*Tg(myofiber:iCre#1)*; see Methods], we readily detected about 50 distinct hues in *palmuscle-Multi*

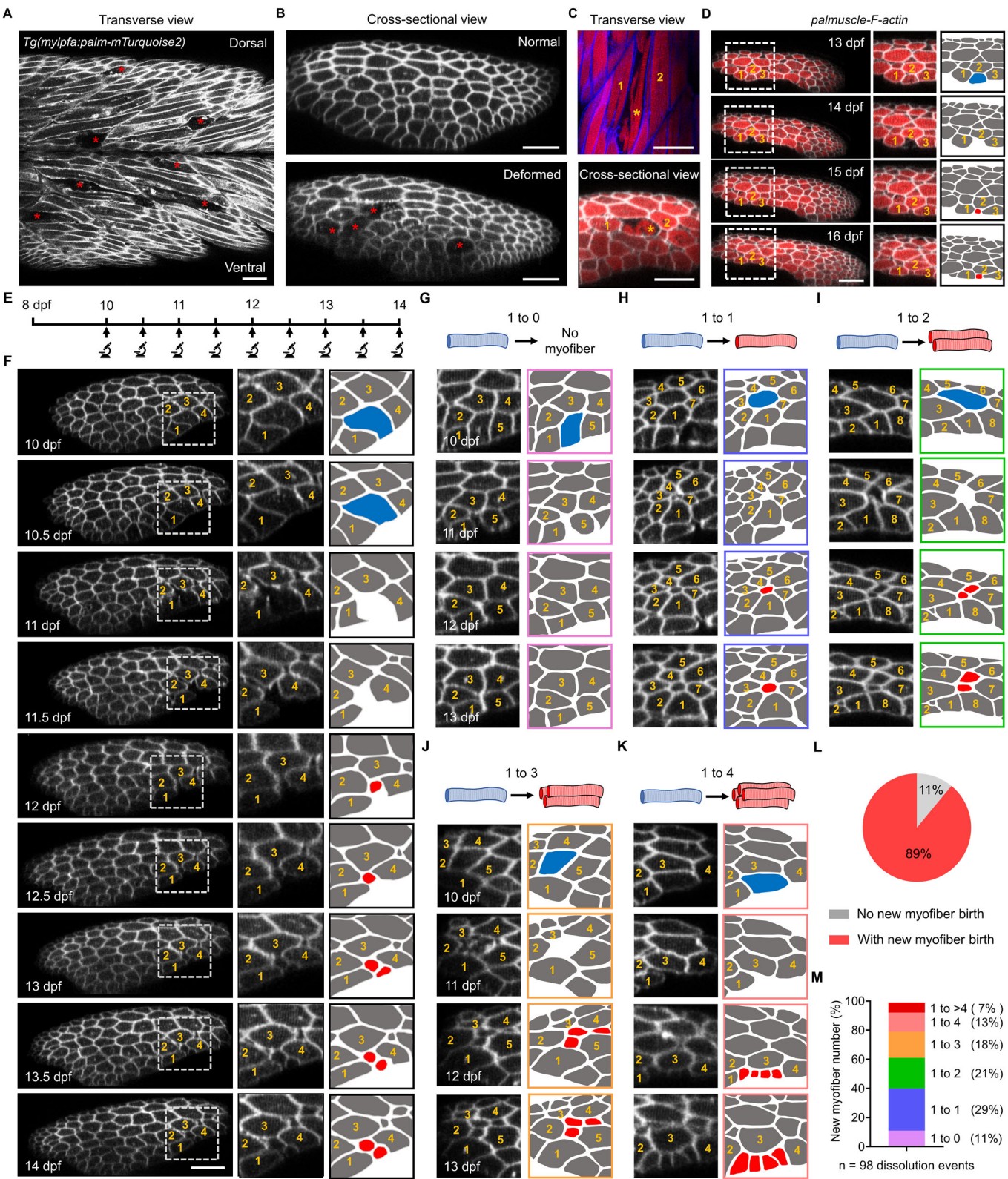

after Cre activation (Fig. 4F). Altogether, the data led us to conclude that *palmuscle-Multi* can generate sufficiently diverse colors for *en masse* barcoding of live myofibers on a whole-animal scale.

Although *Tg(myofiber:iCre#1)* effectively lights up most of the myofibers in *palmuscle-Multi*, we noticed sporadic leakiness in this particular Cre-driver even in the absence of Tamoxifen treatment. To ensure developmental stage-specific labeling of

◄ **Figure 3. Same myofiber tracking in vivo identifies *quid pro quo* myofiber birth.**

(A) Transverse view of the trunk region in the *Tg(mylpfa:palm-mTurquoise2)* line. Red asterisks mark deformed myofibers. (B) Cross-sectional view of a myotome with either ordinary myofibers (top) or deformed myofibers (bottom). Red asterisks mark deformed myofibers. (C) Transverse (top) and cross-sectional (bottom) views of normal and deformed myofibers in *palmuscle-F-actin: Tg(mylpfa:palm-mTurquoise2; mylpfa:LifeAct-mScarlet)*, showing disorganized actin cytoskeleton in a deformed myofiber (yellow asterisk). Neighboring myofibers are labeled with respective numbers. (D) Time-lapse images of the same myotome over a 3-day period, showing a myofiber dissolution event. Magnified view of the region indicated on the left by a white dashed box (middle). Schematic drawing of the magnified region (right) highlights the deformed myofibers (blue) and the newborn myofibers (red). Neighboring myofibers are labeled with respective numbers. (E) Timeline of the tracking scheme. (F) Time-lapse images of the same myotome over a 4-day period at a 12-h interval (left). Magnified view of the region indicated on the left by a white dashed box (middle). Schematic drawing of the magnified region (right) highlights both the deformed myofibers (blue) and the newborn myofibers (red). Neighboring myofibers are labeled with respective numbers. (G–K) Representative time-lapse images of the different dissolution-replacement events. The removal of a single deformed myofiber is spatiotemporally coupled with either no myofiber birth (G), or the birth of either one (H), two (I), three (J), or four myofibers (K). (L, M) All 98 captured dissolution events were categorized as being coupled with no or new myofiber birth (L), and by the number of new myofibers (M). Scale bar, 50 μm (A–D, F). dpf, days post-fertilization. Source data are available online for this figure.

larval myofiber populations upon Cre activation, we implemented a dually inducible system (see Methods for details; Fig. 4G–I), and conducted an additional screening for inducible Cre driver lines that exhibit no leakage. We later identified one such Cre line, *Tg(myofiber:iCre#2)* that showed no leaky Cre activity from head to tail when used in conjunction with *palmuscle-Multi* (Fig. 4J; not a single labeled myofiber was detected among 16 animals). Yet, following a short pulse of doxycycline (Dox) and tamoxifen (Tam), we readily identified many multicolor-barcoded myofibers spanning various anatomical regions (Fig. 4K) Thus, we concluded that, although fewer myofibers may be labeled in each animal, the utilization of *Tg(palmuscle-Multi; myofiber:iCre#2)* enables faithful, long-term monitoring of stage-specific larval myofiber populations throughout various body regions.

## Most larval myofibers are eliminated by adult stages

To determine the extent of the myofiber elimination that may occur during post-embryonic growth, we transiently induced Cre activity in *Tg(palmuscle-Multi; myofiber:iCre#2)* at 4 dpf and then tracked each of the barcoded myofibers at later stages (Fig. 5A–J). To gain a general understanding of how widespread the elimination phenomenon might be, we tracked individual myofibers within three distinct muscle populations of the same animal; the populations included pectoral fin myofibers, craniofacial myofibers, and trunk myofibers. Remarkably, we first observed that each and every labeled myofiber in the 14 dpf pectoral fin is fully eliminated by 28 dpf (100%; *n* = 328 from 12 animals; Fig. 5B,E). Our observations of craniofacial myofibers showed that as many as 54% are eliminated by 28 dpf; after this time, the head region develops strong autofluorescence that hinders accurate tracking (*n* = 138 from 6 animals; Fig. 5C,F). Of note, we detected no leaky Cre activity or any spontaneously labeled myofibers throughout the time period (Fig. EV3A–D). As for the trunk myofibers, we similarly detected elimination in live animals by 21 dpf (Fig. 5D). However, the rapid thickening of the trunk skin during juvenile stages posed an obstacle to live imaging. To overcome this, we continued tracking the tagged myofibers at 42, 70 and 180 dpf through a series of high-content histological examinations at the resolution of single myofibers. Intriguingly, we determined that up to 80% of the larval myofibers were eliminated in full by adult stages (180 dpf/ 6-month-old animals; Fig. 5G–J). We noticed that a small number of the tagged myofibers may persist into adulthood, yet this

population constitutes as little as 0.14% of the adult muscle compartment (Fig. 5J). Consistent with these findings, when we extended the Cre activation period to label up to 52% of the trunk myofibers at 29 dpf, our analysis confirmed the expected substantial elimination of larval myofibers by 70 dpf (96% decrease and 0.13% remaining; Fig. 5K–N). To preclude the possibility of potential silencing of the reporter *palmuscle-Multi* at later stages, which could result in the loss of myofiber signals, we examined reporter expression by inducing Cre activation at three distinct time points (i.e., 18 dpf, 32 dpf, and 10 mpf; Fig. 5O). The results consistently demonstrated robust labeling of myofibers upon Cre activation, assuring that the reporter maintains steady expression levels throughout larval, juvenile, and adult stages (Fig. 5P,Q). Of note, we detected zero tagged myofibers in the controls across all the *Tg(palmuscle-Multi; myofiber:iCre#2)* animals examined at different developmental stages (no treatment of Dox and Tam; 14, 29, 42, 43, 70, 180 dpf, 10 mpf, and 16 mpf; Figs. 4J, 5H,I,L,M,P, 6J,K, EV3C,D and EV3G; *n* = 82 animals), a crucial prerequisite for reliable long-term cell fate mapping.

To independently access the extent and rate of the larval myofiber loss on a whole-animal scale, we further developed a qPCR-based method specifically designed to detect the presence of recombined Brainbow cassette-resulting RNA transcripts from entire animals (see Methods and Fig. 5R–T'). Of note, we examined RNA transcripts as a proxy, as the recombined Brainbow cassette does not produce consistent fixed-length DNA products suitable for qPCR detection. Consistent with the imaging-based findings, we identified a drastic 118-fold increase in transcripts from the recombined cassettes following treatment with Dox and Tam. Over time, the transcript levels progressively decreased, and a 98% reduction in transcripts was seen by 42 dpf (i.e., ~25% reduction per week; FG condition; Fig. 5T). It is worth noting that we examined the remaining recombined cassettes at 70 dpf in three separate body compartments—Anterior (A), Middle (M), and Posterior (P)—owing to the substantial size of the fish at this stage. Intriguingly, yet in line with the histological examination, we detected near-background readouts from all three body compartments (an average of 99% reduction; Fig. 5T,T'). Despite the method being unavoidably influenced by the dilution effect from natural animal growth, the findings affirm the near-complete absence of the recombined cassettes on a whole-animal scale. By synthesizing evidence from live tracking of individual myofibers, histological examination, and in toto detection of the recombined cassettes, we conclude that the majority of larval myofibers exist at

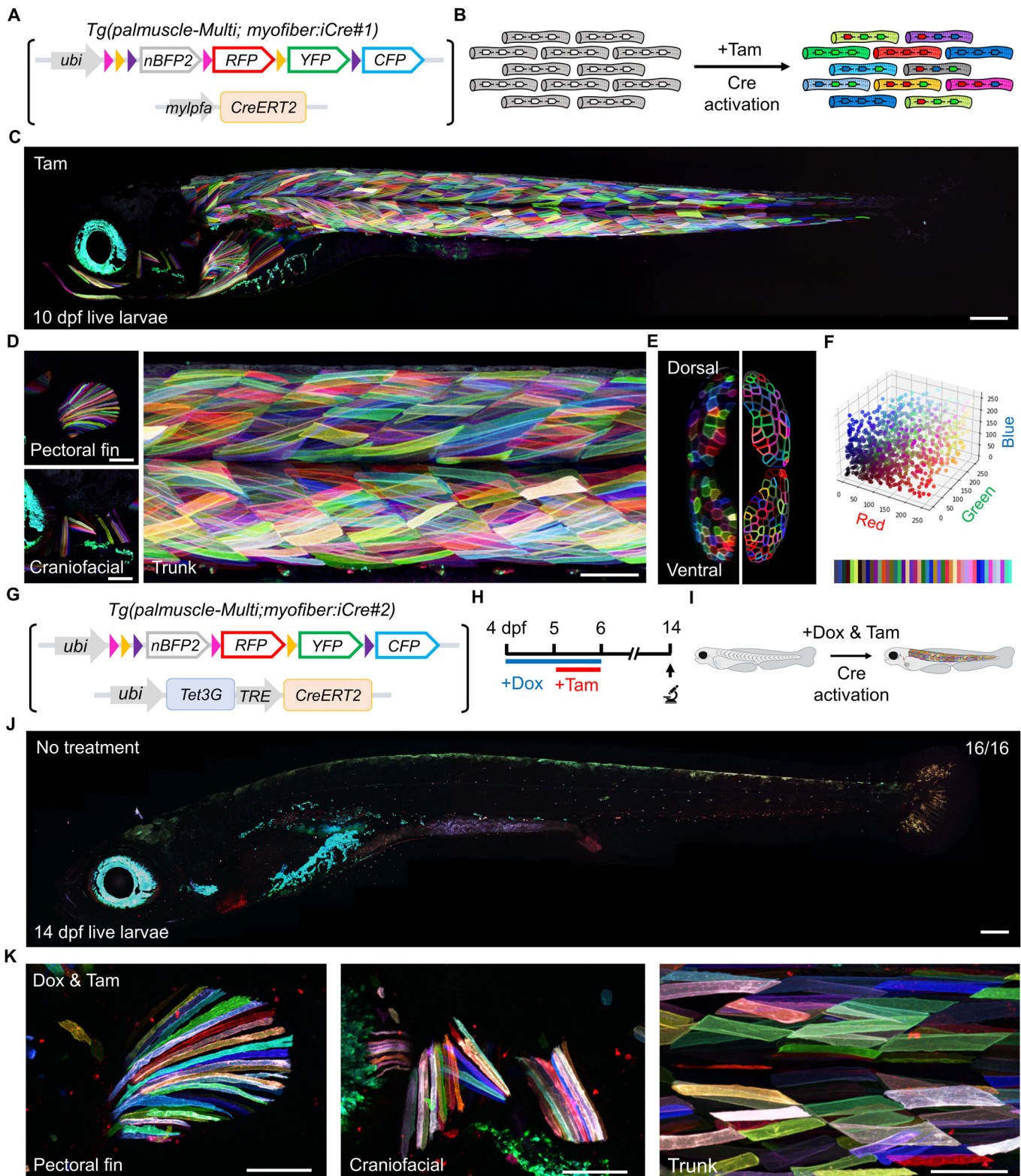

negligible levels as an individual reaches adulthood, irrespective of anatomical position.

To better capture myofiber dynamics during the elimination and replacement processes, we created an additional Brainbow line,

*Tg(mylpfa:Brainbow1.0L)*, which allows for simultaneous monitoring of both pre-existing and de novo populations within the same muscle compartment of the same individual (Fig. EV3E). The design labels all myofibers with tdTomato by default. Thus, the reporter marks all

◄ **Figure 4.  Multicolor barcoding of the entire myofiber population in *palmuscle-Multi* zebrafish.**

(A) The *palmuscle-Multi* and *myofiber:iCre#1* transgenic constructs. (B) Schematic drawing of individual myofibers before and after Cre activation. Addition of tamoxifen (Tam) activates Cre recombinase, which acts on Brainbow-based cassettes to convert label-free myofibers into color-barcoded myofibers. (C) Whole-animal view of a live *palmuscle-Multi* zebrafish larva at 10 dpf. (D) Magnified view of the pectoral fin, craniofacial, and trunk myofibers. (E) Representative cross-sectional view of a myotome (left). Schematic outlines of color-barcoded myofibers are shown on the right. (F) Color space analysis of 2057 individual myofibers from 32 myotomes captured from a single *palmuscle-Multi*. About 50 distinct hues were detected in live animals upon Cre activation. (G) The *palmuscle-Multi* and *myofiber:iCre#2* transgenic constructs. (H) Timeline of the treatment and tracking scheme. (I) Schematic drawing of the *Tg(palmuscle-Multi; myofiber:iCre#2)* larva before and after Cre activation. (J) Whole-animal view of the *Tg(palmuscle-Multi; myofiber:iCre#2)* larva without Dox and Tam treatment. (K) The pectoral fin, craniofacial and trunk region displayed multicolor myofibers upon treatment with Dox and Tam. $n$ = number of animals (J). Stitched image (C, D, J). Scale bars, 200 μm (C, J); 100 μm (D, K). dpf, days post-fertilization. Source data are available online for this figure.

myofibers regardless of their date of birth. Without Dox and Tam treatment, we detected no leaky Cre activity in the double transgenic animals, *Tg(mylpfa:Brainbow1.0L; myofiber:iCre#2)*. All pectoral fin myofibers in the control animals remained mono-colored across all examined time points (Fig. EV3F,G). To monitor the interaction between pre-existing and new myofibers, we first labeled the larval myofibers at 14 dpf through Dox and Tam treatment. Subsequently, we conducted same-animal tracking of the entire pectoral fin myofiber population at 21 and 28 dpf. Notably, yet consistent with the findings in *palmuscle-Multi* (Fig. 5), we determined that all of the tagged larval myofibers were eliminated by 28 dpf (100%; $n$ = 43 myofibers in 3 animals; Fig. EV3H,I). The void spaces were readily occupied by new myofibers, which were generally more numerous and larger in size (increases of 233% and 76%, respectively; Fig. EV3J–L). This replacement presumably occurred through the *quid pro quo* myofiber birth mechanism (Fig. 3). Thus, in the pectoral fin compartment, all larval myofibers great and small have a limited lifespan. We therefore propose that the skeletal myofiber populations at adult stages are built almost entirely from scratch during the post-embryonic growth period, with little or no contribution from the larval pool.

## Muscle growth demands regulate myofiber elimination

To determine whether large-scale myofiber elimination is an adaptable mechanism during post-embryonic muscle growth, we manipulated zebrafish larval growth rates as in the previous experiments (SG vs. FG; Fig. 2). Compared to animals kept under the SG condition, zebrafish in the FG condition exhibited drastically enhanced animal growth by 28 dpf, as determined by standard length and trunk surface area (19% and 57%, respectively; Fig. EV4A–C). Intriguingly, we found that only about 55% of the myofibers were eliminated in fish kept in the SG condition ($n$ = 235 myofibers in 10 animals; Fig. EV4D,F). In contrast, nearly all of the pre-existing, pectoral fin myofibers were eliminated by 28 dpf when animals were grown at the faster pace (99% on average; $n$ = 109 myofibers in 5 animals; standard length more than 9 mm; Fig. EV4E,F). We also noticed that the overall animal growth under the FG condition was relatively variable, and several individuals remained small (i.e., standard length less than 9 mm by 28 dpf). Notably, all of the smaller animals retained more myofibers than their larger-sized siblings and neighbors (49% on average; $n$ = 126 myofibers in 5 animals; Fig. EV4E–G). Furthermore, the growth effect on myofiber elimination was not confined solely to the pectoral fin myofibers; rather, it extended to the entire myofiber population, as evidenced by the qPCR-based assay of the recombined Brainbow cassettes from whole animals (63% reduction at 42 dpf; Fig. EV4H). Taken together, these findings led us to

conclude that the elimination process eventually impacts all of the myofibers, but the rate of the elimination is not fixed. Instead, the rate can be intricately adjusted to reflect muscle demands during post-embryonic growth, which can vary widely among individuals.

## Larval myofibers and adult myofibers are two distinct populations

As most of the larval myofibers are either eliminated or replaced by adult stages, we suspected that this large-scale replacement must serve a practical purpose, somehow fulfilling the needs of adult animals. To identify differences between the larval and later myofiber populations, we inspected individual trunk myofibers collected from three key stages of the development (Fig. 6A): 14 dpf (larval stage at which most myofibers are larval myofibers), 70 dpf (juvenile stage at which most myofibers are replacement myofibers), and 1.7 years of age (late adult stage with adult myofiber populations). Intriguingly, we determined that the 70 dpf myofibers were on average 299% longer and 604% larger, with 469% more nuclei than the 14 dpf myofibers ($n$ = 295 vs. 259 myofibers; Fig. 6B–E). Overall, the 70 dpf and adult myofiber populations had much higher morphological similarity to each other than either population did to the 14 dpf myofibers (Fig. 6C–E). In contrast to larval myofibers, adult myofibers demonstrate elevated expression levels of the intermediate filament *desma* and the sarcomeric myosin *myhc4* (increases of 398% and 214%, respectively; Fig. EV5A–C), indicative of major differences in muscle structure and function (Paulin and Li, 2004; Schiaffino et al, 2015). Consistent with these findings, we also determined that adult myofibers display markedly increased mitochondrial respiration capacity and glycolytic activity, as assessed by spare respiratory capacity (SRC) and extracellular acidification rate (ECAR) (respective increases of 77% and 112%; Fig. 6F–H).

To determine whether whole-body myofiber elimination also occurs at adult stages, we transiently induced Cre activity in *Tg(palmuscle-Multi; myofiber:iCre#2)* at 10-month-old fish to specifically label 'late-comer' adult myofibers. The long-term fates of these late-comer myofibers were then evaluated 6 months later (Fig. 6I). It is important to note that the double transgenic line showed no sign of leaky Cre activity at either 10 or 16 mpf (no tagged myofibers detected in the entire section; $n$ = 9 animals; Fig. 6J,K); yet, a short pulse of Dox and Tam effectively labeled ~89% of the 10 mpf adult myofibers ($n$ = 3 animals; Fig. 6J,L). Intriguingly, although the tagged myofibers were mostly near the periphery and the occupied percent area appeared much reduced (~43% less; Fig. 6J,L), we found that the absolute number of tagged myofibers in an animal had little or no change throughout the

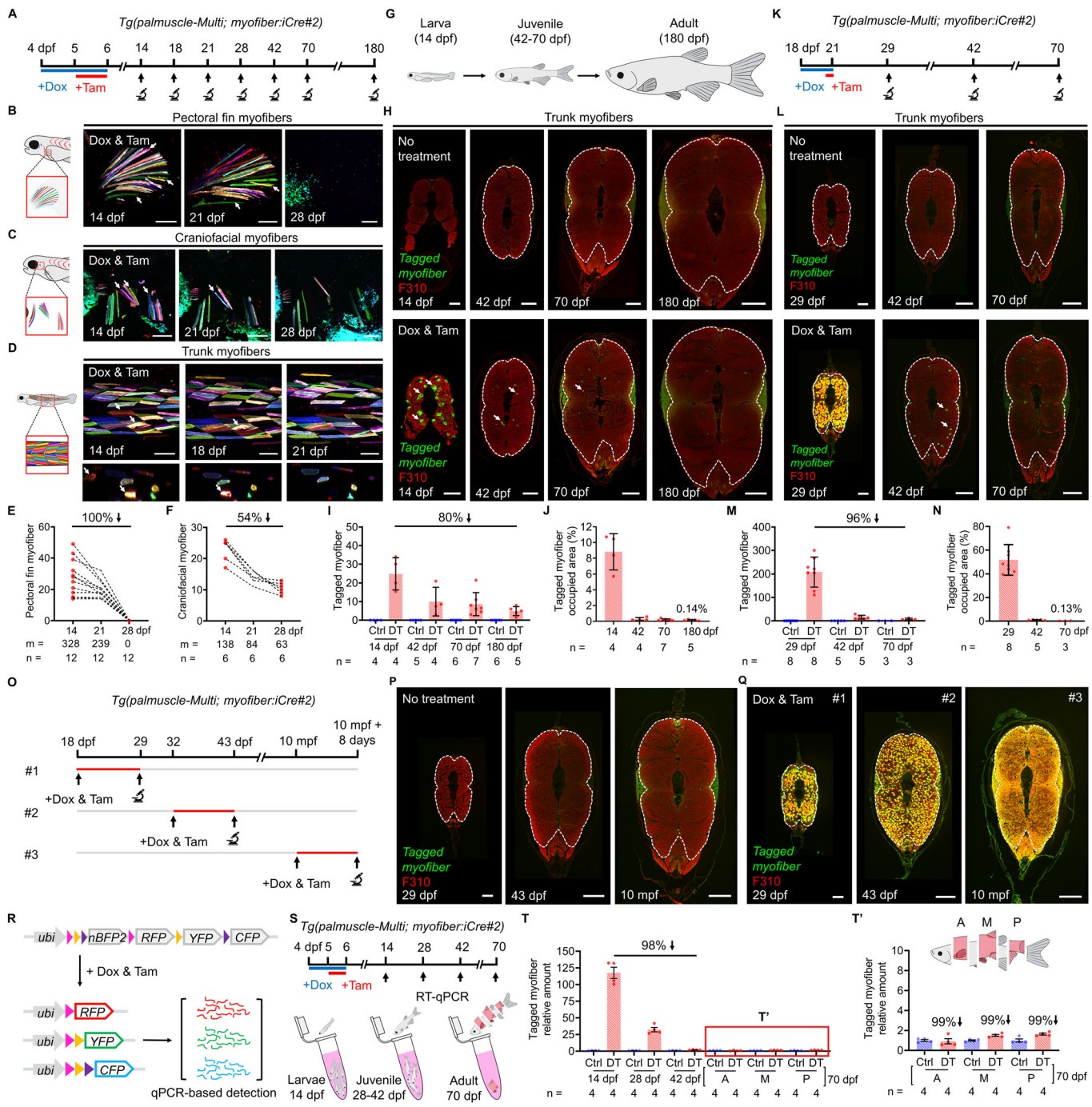

entire 6-month period (3428 ± 864 vs. 3169 ± 256 per cross-section; $n = 3$ animals each; Figs. 6K and EV5D–F). Thus, adult myofibers are not a transient population but rather appear to be long-lived. Of note, we fortuitously captured a progressive shift in the zone of hyperplastic growth during the post-embryonic growth periods. Specifically, the zone transitioned from the periphery at 29 dpf as previously described (Nguyen et al, 2017), to the interstitial space at 43 dpf, and ultimately to the deep region of the myotome by the time the individuals reached 16 months of age (Fig. EV5G–K). Taking our results of cell sizes, shapes, nuclear numbers, molecular

compositions, energy metabolism profiles, and lifespans altogether, we conclude that larval and adult myofibers are two distinct entities. Unlike larval myofibers, adult myofibers are built to last with physical and physiological properties appropriate for adult stage-specific muscle requirements.

## Autophagic cell death mediates myofiber elimination

As skeletal muscles play a vital role in the locomotion and movement of animals, we speculated such large-scale elimination of

myofibers must be tightly regulated. After conducting a literature search for specific molecular mechanisms that could potentially mediate the elimination process, we first hypothesized that apoptosis machinery may be involved (de Torres et al, 2002; Meier et al, 2000; Sandri and Carraro, 1999). To determine whether apoptotic signaling has any role in the elimination process, we created a stable transgenic line, *Tg(mylpfa:H2A-mCherry-2A-GC3AI)*, for live monitoring of caspase-3/7 activity (Zhang et al, 2013) in the larval myofiber populations (Fig. 7A). The reporter functioned as expected upon application of an external apoptotic stimulus (short-term 12-h treatment of cycloheximide; Fig. 7B), yet we failed to detect any activation of the fluorescent reporter from a survey of 9678 individual myofibers, irrespective of whether the myofiber appearance was normal or deformed ($n = 49$ myofibers from 7 animals; Fig. 7C,D). Thus, we concluded that apoptosis is unlikely to account for the elimination. Next, we wondered whether a less-understood cell death mechanism, autophagic cell death (ACD) (Jung et al, 2020; Noguchi et al, 2020), could play a role in the process. To determine whether autophagic responses are elevated in the deformed myofiber population, we first applied LysoTracker to detect autolysosome formation in live zebrafish larvae (He and Klionsky, 2010). Intriguingly, we found that as many as 80% of the deformed myofibers contained red puncta in the cytoplasm ($n = 33/41$ myofibers from 11 animals; Fig. 7E–G), which is indicative of excessive autophagic activation (i.e., formation of acidic autolysosomes). In contrast, only about 0.1% of the adjacent, non-deformed myofibers contained red puncta ($n = 3/2728$ myofibers from 11 animals; Fig. 7F,G). To better visualize autophagic activation in the myofibers, we cloned zebrafish LC3 (mammalian homolog of Autophagy-related protein 8) and created a reporter construct that constitutively expresses mCherry-GFP-LC3 in the myofiber populations (*mylpfa:mCherry-EGFP-LC3*). Upon autophagic activation, yellow puncta-containing autophagosomes will fuse with lysosomes to produce red puncta-containing autolysosomes due to the acid sensitivity of EGFP (Fig. EV6A) (Pankiv et al, 2007). Of note, we validated the reporter function by treating animals with the autophagy activator rapamycin (Fig. EV6B). Consistent with the findings using

Figure 5. Most larval myofibers are eliminated by adult stages.

LysoTracker, we saw that the deformed myofibers contained high ratios of red puncta to yellow puncta (79% to 100%; $n = 146$ puncta from 3 myofibers; Fig. EV6C,D), indicating autophagic activation. Furthermore, we found that a 48-h treatment of the autophagy inhibitor chloroquine (CQ) could effectively prevent full elimination of pectoral fin myofibers by 28 dpf (96% vs. 77% decrease in labeled myofibers; $n = 182$ vs. 284 myofibers; Fig. 7H–K). Since animal growth rates can also affect the extent of the myofiber elimination (Fig. EV4), we limited this experiment to include only individuals with standard length above 9 mm (Fig. 7J). In line with these findings, we further established that the CRISPR/Cas9-mediated genetic knockdown of the master autophagy regulator ATG7 (resulting in a 72% reduction of ATG7 expression as determined by RT-qPCR; Fig. 7L–N) had a marked impact on the extent of elimination (92% vs. 42% decrease in labeled myofibers; $n = 249$ vs. 289 myofibers; Fig. 7O–R). From these findings, we concluded that the elimination of larval myofibers aligns with the key criteria of ACD (Shen and Codogno, 2011): (1) no activation of typical caspase-dependent cell death pathways; (2) widespread presence of acidic autophagic vesicles; (3) the effectiveness of both pharmacological and genetic interventions in preventing cell death resulting from autophagic activation. Thus, we propose that ACD may act as a decisive mechanism for the orderly removal of the larval myofibers during post-embryonic growth.

## Autophagic cell death triggers local addition of new myofibers

To determine whether ACD may affect myofiber birth at the voided site, we created an inducible cassette, *mylpfa:Tet3G/TRE:EGFP-2A-ATG7*, for driving ATG7 expression in a spatiotemporally controlled manner (Fig. 7S). Since ATG overexpression alone is sufficient to elevate autophagic responses in multiple models (Garcia-Prat et al, 2016; Pattison et al, 2011; Pyo et al, 2013) and key autophagy genes (i.e., *atg5* and *atg7*) (Dikic and Elazar, 2018) were highly expressed in larval myofibers (Fig. EV6E,F), we hypothesized that forced, prolonged overexpression of ATG7 in larval myofibers may activate ACD in a synchronized manner. To

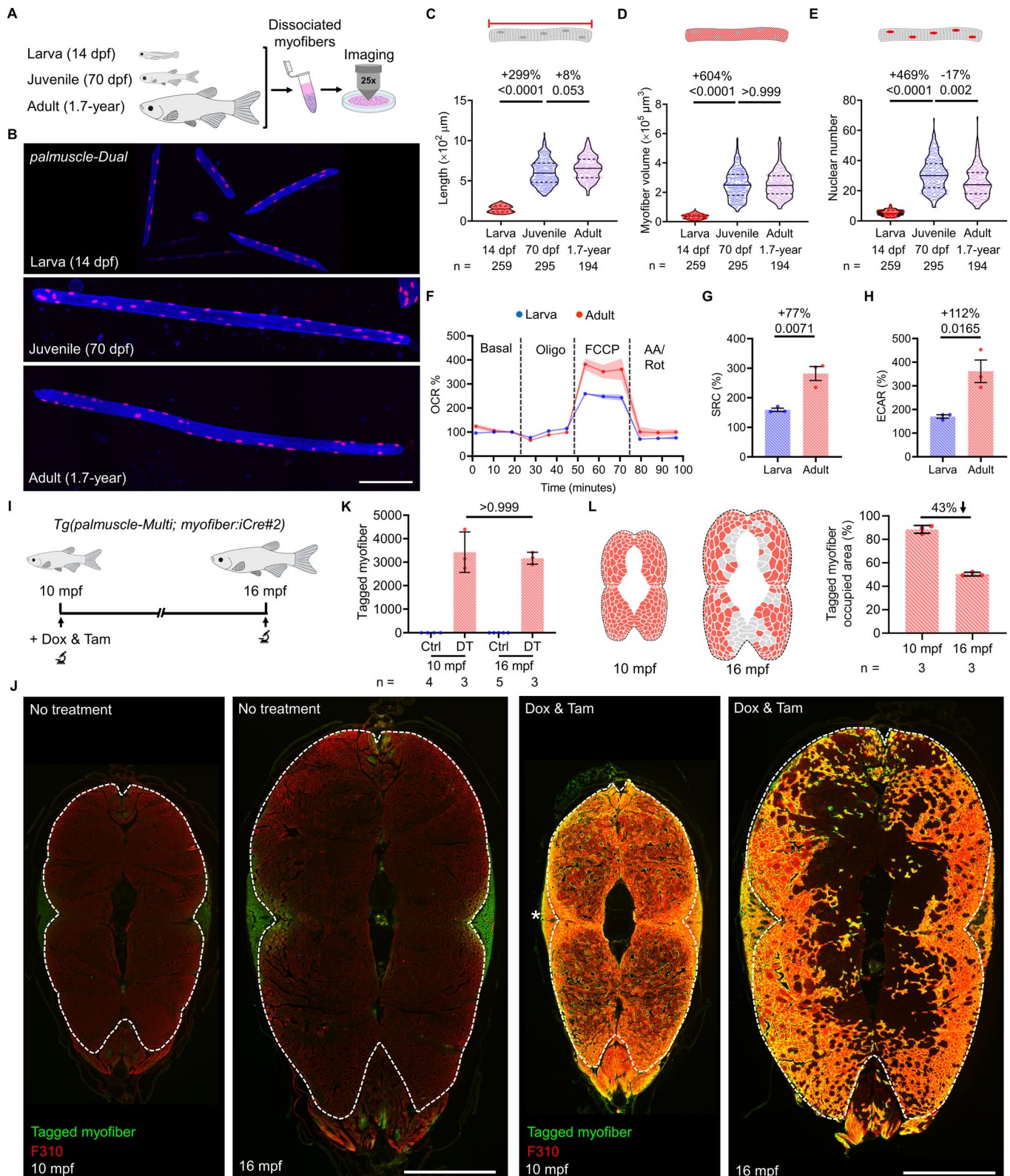

test this possibility, we first determined that Dox treatment is effective to induce EGFP expression in the myofibers (Fig. 7T,U). Intriguingly, ATG7 overexpression starting at 9 dpf potently triggered *en masse* elimination of EGFP-positive larval myofibers

by 14 dpf (Fig. 7U; right panels). Strikingly, upon release of the Dox treatment, the extreme ATG7-triggered loss of myofibers led to booming production of new myofibers in situ. The vacant intercellular spaces were filled by large clusters of newborn

◄ **Figure 6. Larval myofibers are distinct from their adult counterparts.**

(A) Illustrations of zebrafish at larval, juvenile and adult stages reflect relative size differences; dissociated myofibers were subjected to imaging analysis. (B) Representative confocal images of dissociated myofibers collected from *palmuscle-Dual* at 14 dpf, 70 dpf, and 1.7 years of age. (C–E) Quantification of myofiber length (C), volume (D), and total nuclear number within each myofiber (E). (F) Real-time monitoring of oxygen consumption rate (OCR; percentage) from freshly dissociated larval and adult myofibers. Mitochondrial stress modulators: oligomycin (oligo), FCCP, and antimycin A/rotenone (AA/Rot). Solid lines indicate mean and colored shadows highlight standard error. (G, H) Quantitative changes in spare respiratory capacity (SRC; percentage) and extracellular acidification rate (ECAR; percentage). (I) Timeline of the treatment and tracking schemes. (J) Histological examinations of the tagged myofibers in the middle-trunk region of animals at 10 and 16 mpf. White dashed line encircles the area containing fast muscle fibers. Of note, *Tg(myofiber:iCre#2)* labeled both fast and slow myofiber populations at adult stages. White asterisks highlight the slow myofiber domains. F310 Ab stains fast myofibers. (K, L) Quantification of tagged myofiber number (K) and percent area occupied by the tagged myofibers (L). The tagged myofiber numbers from three consecutive cross-sections were counted and averaged for each individual. Of note, comparisons were made between animals with at least 10% of the labeled area per cross-section. Data from biological replicates are shown as violin plots (solid lines, median; dashed lines, quartiles; C–E), mean ± standard error (G, H) and mean ± standard deviation (K, L). Significance was examined by Kruskal–Wallis test with Dunnett's correction (C–E); two-tailed Student's *t*-test (G, H) or two-tailed Mann–Whitney test (K). Percent differences and *P* values are shown above the horizontal lines for intergroup comparisons. *n* = number of myofibers (C–E) or animals (K, L). Stitched image (B, J). Scale bars, 100 μm (B); 1 mm (J). dpf, days post-fertilization. mpf, months post-fertilization. Source data are available online for this figure.

myofibers within 3 days (*n* = 12 myofibers per cluster on average; *n* = 66 clusters; Fig. 7U–W). Of note, the extreme myofiber loss induced in this experiment may not naturally occur during animal growth. We detected no such 'myofiber clusters' in any of the control 17 dpf animals with only EGFP overexpression (*n* = 93 vs. 78 myotomes in 5 to 6 animals; Fig. 7U,W) or during normal development (Fig. 3). Taken together, these findings suggest a tight coupling between ACD and the *quid pro quo* myofiber birth events. Thus, it appears that ACD mediates the timely loss of larval myofibers, and the coupling mechanism ensures an efficient on-demand filling of the vacated space with new myofibers that are often greater in number.

## Discussion

Post-embryonic muscle growth in vertebrate animals has long been thought to proceed by two modes, an increase in myofiber number (hyperplasia) and an expansion in myofiber volume (hypertrophy). Here, we performed in toto monitoring of individual myofibers throughout larval, juvenile and adult stages, which allowed us to identify a distinct mode of muscle growth that occurs alongside hyperplasia and hypertrophy and impacts nearly all of the larval myofibers (Fig. 7X). As the whole-body replacement of larval myofibers proceeds at a gradual yet flexible pace, animals may experience no adverse effects during the wholesale swap of the body's most abundant tissue. Thus, there may seem to be no drastic change in the physical appearance of the animals during the post-embryonic growth period, but the internal skeletal muscle tissues are entirely reconstructed piece-by-piece in order to prepare the individual for its adult stage.

As skeletal myofibers have the ability to grow in size through hypertrophy, why is it necessary for the replacement to occur during the larva-to-adult transition phase? We speculate that myofibers that had been customized with distinctive characteristics (e.g., architectures, sizes, shapes, and functions) for specific physiological needs of animals at the larval stages have limited growth potential. Instead of going through a series of de-differentiation and re-differentiation programs to reshape the existing populations, the replacement mechanism may thus represent a dramatic yet more efficient means to generate a full set of myofibers that fit with adult-specific needs. Consistent with this notion, we determined that (1) the size growth of larval

myofibers is temporally constrained (Fig. 2K); (2) the growth rate of the animals determines the elimination speed (Fig. EV4); (3) the adult myofibers derived from replacement are morphologically and functionally distinct from larval myofibers (Figs. 6 and EV5A–C). Thus, reminiscent of the wholesale replacement of human baby teeth with adult teeth, we propose that extensive replacement but not incremental growth is an economically efficient strategy for transforming complex tissues and organs that had been previously tailored for a particular developmental stage. Meanwhile, it is important to note that zebrafish can undergo significant growth during the larva-to-adult transition phase, which is expected to reduce the proportion of larval myofibers even in the absence of the replacement. However, we have provided three independent lines of evidence to support the extensive elimination of larval myofibers: 1) continuous live tracking of approximately 500 individual myofibers across two different transgenic lines (Figs. 5A–F and EV3E–I), (2) millimeter-scaled histological examination of the tagged myofibers at single-cell resolution (Figs. 5G–N and EV5D–F), and (3) whole-body qPCR-based detection of the recombined cassettes (Fig. 5R–T'). Altogether, these coherent findings support the occurrence of extensive replacement within the pre-existing larval myofiber populations.

In this study, we identified four essential features of larval myofiber replacement: *quid pro quo* birth, characteristic developmental timing, growth adaptability, and ACD activation. Intriguingly, ACD (not apoptosis) was previously shown to be the primary mechanism of removing obsolete larval midgut cells and salivary glands during fly metamorphosis (Berry and Baehrecke, 2007; Denton et al, 2009). Future studies are therefore needed to determine whether ACD is an evolutionarily preferred mechanism for directing wholesale tissue replacement during animal transitions to adulthood (Allen and Baehrecke, 2020; Denton and Kumar, 2019; Jung et al, 2020). More work will also be required to learn how ACD is triggered and managed in the zebrafish skeletal muscle such that the elimination-replacement process removes just the right amount of in-service myofibers at any given time without compromising the vital body functions of the animals. It also remains to be determined whether dying myofibers may release any specific metabolites and/or cytokines (Medina et al, 2020; Ratnayake et al, 2021) to promote myofiber births in situ, how the muscle stem cells and clonal growth may be stimulated (Gurevich et al, 2016; Hughes et al, 2022; Nguyen et al, 2017), and what crucial factors impose a size limit on larval myofiber growth in the first place.

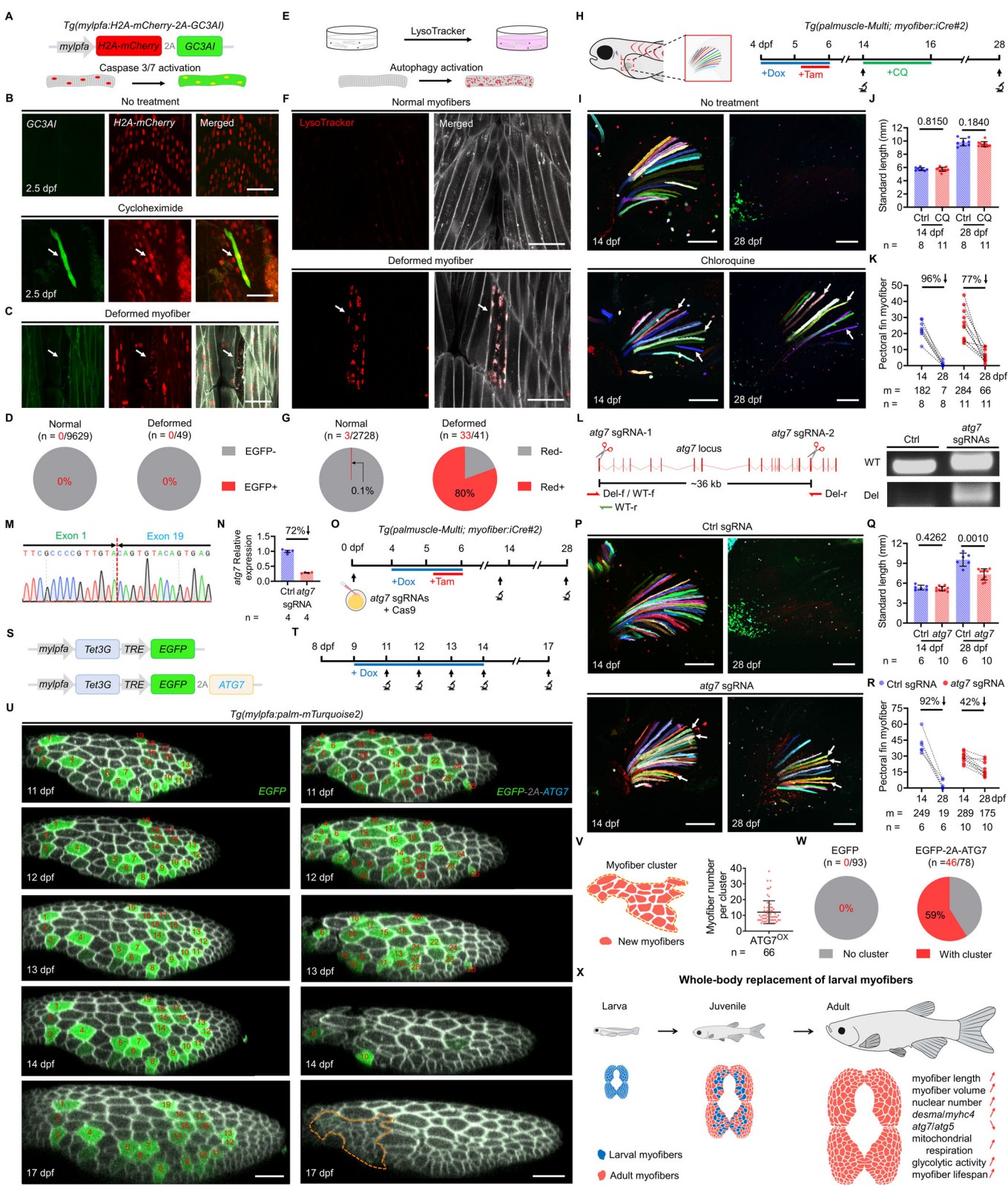

Considering the correlation of thyroid hormone (TH) with the metamorphosis-related tail muscle death in Xenopus (Das et al, 2002) and adult transitions in zebrafish (McMenamin et al, 2014), we postulated that TH may similarly act as a trigger for post-embryonic

muscle death in zebrafish. However, our experiments using three treatment schemes and two different concentrations (Brown, 1997; Porazzi et al, 2009) did not reveal any notable effect of TH on the extent of larval myofiber loss (Fig. EV6G–J). While we cannot entirely

**Figure 7. Autophagic cell death eliminates larval myofibers to promote local myofiber birth.**

(A) The transgenic construct for live monitoring of apoptotic responses in the myofibers. (B) Treatment with cycloheximide activates apoptosis in the myofibers. White arrow points to an EGFP-positive myofiber. (C) Deformed myofibers show no EGFP signals. White arrow points to a deformed myofiber. (D) Quantification of the EGFP-positive myofibers either in the normal or the deformed populations. (E) The application of LysoTracker red for detecting autophagic activation in the myofibers. (F) Deformed myofibers contain the LysoTracker red signals. White arrow points to a deformed myofiber. (G) Quantification of the LysoTracker red-positive myofibers either in the normal or the deformed populations. (H) Timeline of the treatment and tracking schemes. (I) Treatment of chloroquine (CQ) attenuates myofiber elimination. White arrows point to persisting myofibers. (J, K) Standard length of the animals (J) and quantitative evaluation of tagged myofiber number (K). Since animal growth rates can affect the myofiber elimination, the analyses only include larvae with standard length more than 9 mm at 28 dpf. (L, M) CRISPR-Cas9-mediated deletion of the *atg7* locus and evaluation of the deletion alleles by PCR amplification of genomic DNA (L) and Sanger sequencing (M). The sgRNA pair removes the ~36 kb genomic region between Exon 1 and Exon 19. (N) RT-qPCR analysis of CRISPR-Cas9-mediated knockdown of the *atg7* expression. (O) Timeline of the treatment and tracking schemes. (P) Pectoral fin myofibers images from 14 dpf larvae injected with either control sgRNAs or *atg7*-targeting sgRNAs. White arrows point to persisting myofibers. (Q, R) Standard lengths of the animals (Q). Quantitative evaluation of tagged myofiber number (R). Of note, animals injected with sgRNAs targeting *atg7* exhibited reduced growth by 28 dpf. (S) The transgenic constructs for inducible overexpression of ATG7. (T) Timeline of the Dox treatment and tracking schemes. (U) Time-lapse images of the same myotome over a 6-day period with overexpression of either EGFP (left) or EGFP/ATG7 (right). The EGFP and EGFP/ATG7-positive myofibers are marked with respective numbers. Orange dashed line highlights a myofiber cluster. (V) Quantification of the myofiber number within each cluster. Of note, concurrent births of five or more than five adjacent myofibers were defined as a myofiber cluster. OX, overexpression. (W) Quantitative myofiber cluster appearance in the EGFP- and EGFP/ATG7-overexpression groups. (X) Schematic drawing depicts the process of whole-body replacement of larval myofibers. Data from biological replicates are shown as mean ± standard deviation (J, Q, V) or mean ± standard error (N). Significance was examined by two-tailed Mann–Whitney test (J, Q) or two-tailed Student's t-test (N). Percent differences and *P* values are shown above the horizontal lines for intergroup comparisons. m = number of myofibers (K, R). *n* = number of animals (J, K, Q, R), myofibers (D, G), clusters (V), or myotomes (W). Scale bar, 50 μm (B, C, F, U); 100 μm (I, P). dpf, days post-fertilization. Source data are available online for this figure.

rule out the possibility that TH is involved in promoting muscle replacement in zebrafish, it is worth noting that the muscle death program in Xenopus involves caspase activity; whereas in zebrafish, we only detected autophagy activation in the dying larval myofibers, and no caspase activity was found (Fig. 7A–G). In addition, our finding that *pax7b*-positive muscle stem cells show enrichment near the deformed myofibers (Fig. EV2F–I) warrants further investigations into whether these cells can undergo extensive clonal growth, as depicted in the seminal study by Nguyen et al (Nguyen et al, 2017). We also made an unexpected observation of a progressive shift in the zone of hyperplastic growth during the post-embryonic growth period in zebrafish (Fig. EV5G–K). Thus, more work is needed to determine whether different types of muscle stem cells participate in this growth phenomenon and whether the whole-body replacement of larval myofibers may have a role in influencing this shift.

In summary, as myogenesis processes are evolutionarily conserved across both amniotes and teleosts (Goody et al, 2017; Manneken et al, 2022) and signs of naturally occurring muscle cell death are commonly observed during normal vertebrate development, including in chick, rat, and humans (de Torres et al, 2002; McClearn et al, 1995; Sandri and Carraro, 1999; Webb, 1972), we speculate that the wholesale swap of early-born myofibers could be a common occurrence in vertebrate models. This process might be relevant in the context of muscle disorders, such as childhood- and/or late-onset muscular dystrophies, and sarcopenia (Bonnemann et al, 2014; McNally and Pytel, 2007; Talbot and Maves, 2016).

## Methods

### Zebrafish and transgenic constructs

A 2.2-kb promoter sequence upstream of the *mylpfa* gene (Ju et al, 2003) was used to generate following transgenic lines: Tg(mylpfa:palm-mTurquoise2)as69, Tg(mylpfa:H2A-mCherry)as70, and Tg(mylpfa:Brainbow1.0L)as71, and Tg(mylpfa:H2A-mCherry-2A-GC3AI)as72, and Tg(mylpfa:LifeAct-mScarlet)as76, the Brainbow reporter sequence was a gift from Joshua Sanes (Addgene plasmid # 18721) (Livet et al, 2007).

The GC3AI reporter sequence was a gift from Binghui Li (Addgene plasmid #78910) (Zhang et al, 2013). The LifeAct-mScarlet reporter sequence was a gift from Dorus Gadella (Addgene plasmid #85056) (Bindels et al, 2017). Tg(ubi:palmbow)as73 or *palmuscle-Multi* was generated with the 3.5-kb *ubi* promoter and the *palmbow* sequence (Loulier et al, 2014). Tg(myofiber:iCre#1) and Tg(myofiber:iCre#2) were generated using the following transgenic constructs: Tg(mylpfa:CreERT2)as74 and Tg(ubi:Tet3G/TRE:CreERT2)as75. In as75, a dually inducible system was included (Chan et al, 2022). Either a *cmlc2:nBFP2* fragment (as74) or a *cmlc2:mCherry* fragment (as75) was co-injected with the Cre expression constructs as a selection marker. Of note, despite being driven by the *ubi* promoter, the Tg(myofiber:iCre#2) line consistently labels the fast myofiber population, and sporadically labels the slow myofiber and superficial skin cell populations upon Dox/Tam treatment. It is worth noting that this line displayed no Cre activity in muscle stem cells, as evidenced by the absence of new myofiber additions across all of our tracking experiments (Figs. 5B,C,H,I,L,M, 6J,K, EV3H and EV4D,E; n = 83 animals). Moreover, throughout our examination of all developmental stages, the line demonstrated no detectable leakiness (14, 29, 42, 43, 70, 180 dpf, 10 mpf, and 16 mpf; Figs. 4J, 5H,I,L,M,P, 6J,K, EV3C,D and EV3G; *n* = 82 animals). The 4.2 kb *pax7b* promoter sequences, the 2.2 kb *atg7* and 369 bp *lc3b* coding sequences were amplified from the wild-type zebrafish genomic DNA and cDNA preparations to generate transgenic constructs of Tg(pax7b:mCherry), Tg(mylpfa:Tet3G-TRE-EGFP-2A-ATG7) and Tg(mylpfa:mCherry-EGFP-LC3b). These three constructs were investigated using mosaic transgenic animals. The primer sequences used for cloning are listed in Table EV1. All transgenesis experiments were facilitated with I-SceI meganuclease. Zebrafish Ekkwill (EK) strain was used to generate the transgenic lines. All transgenic lines and constructs can be obtained from the corresponding author upon request, without any limitations.

### Fish growth, maintenance, and study design

Zebrafish larvae were fed with paramecia starting at 8 dpf, and the rearing density was maintained at 1 animal per 20 ml from 8 to 10 dpf. From 10 to 14 dpf, larvae were fed with increasing amounts of

paramecia. Beyond 14 dpf, larvae were fed with both artemia and solid fish food. Two rearing conditions were used to establish different larval growth rates: 1 animal per 20 ml (slow growth; SG) and 1 animal per 200 ml (fast growth; FG). In most experiments, the fast growth condition was used unless otherwise specified. Adult fish were maintained at 4–5 individuals per liter under an artificial photoperiod of 14:10 h light-dark cycle at 26–28 °C. Sample size was determined based on our prior experience with the zebrafish model. Animals were randomly allocated into control and experimental groups. No animals were excluded from the analysis. The investigators were not blinded to allocation during experiments and data assessment due to the same investigator processing the samples and collecting the data. Thus, blinding was not practical. The Institutional Animal Care and Utilization Committee (IACUC) at Academia Sinica approved animal procedures carried out in this study.

## Cre activation in Tg(myofiber:iCre#1) and Tg(myofiber:iCre#2)

Cre activity was transiently induced in larvae and adult zebrafish with tamoxifen (2 µM; Sigma, T5648) alone or in combination with doxycycline (20 µg/ml; Sigma, D9891). The induction conditions were optimized for each transgene combination: (1) 2-h incubation with tamoxifen for Tg(palmuscle-Multi; myofiber:iCre#1) at 4 dpf; (2) combined incubation of 48-h doxycycline and 18-h tamoxifen for Tg(palmuscle-Multi; myofiber:iCre#2) and Tg(mylpfa:Brainbow1.0L; myofiber:iCre#2) at 4 dpf; (3) combined incubation of 72-h doxycycline and 2-h tamoxifen for Tg(palmuscle-Multi; myofiber:iCre#2) at 18 and 32 dpf; (4) combined incubation of 72-h doxycycline and 4-h tamoxifen for Tg(palmuscle-Multi; myofiber:iCre#2) at 10 mpf. The samples were collected 8 days later after the Dox/Tam treatment.

## Live image acquisition

All images were acquired using a Leica SP8 and Stellaris 8 confocal with 25x water immersion lens (25x/0.95 HCXIRAPO). Animals were anesthetized with tricaine (0.4 mg/mL) and mounted in 1.2% low-melting agar for imaging. Images were acquired with the following settings, unless otherwise specified: (1) format, $1024 \times 1024$ (i.e., 465 µm × 465 µm); (2) speed, 600 Hz in bi-directional mode; (3) pinhole size, 1 or 2 AU, (4) line average, 1 or 2; (5) z-stack, 1 or 2 µm. palmuscle-Multi images were acquired using the following settings: (1) 448 nm laser (CFP), 450- to 510-nm bandpass; (2) 514 nm (YFP), 520- to 550-nm bandpass; (3) 552 nm (RFP), 585- to 649-nm bandpass; (4) pinhole size of 2 AU, line average of 1, and z-stack with 2-µm step size. Tg(mylpfa:Brainbow1.0L) images were acquired using the following settings: (1) 448 nm laser (CFP), 450- to 510-nm bandpass; (2) 514 nm (YFP), 520- to 550-nm bandpass; (3) 552 nm (RFP), 600- to 615-nm bandpass; (4) pinhole size of 2 AU, line average of 1, and z-stack with 2-µm step size. palmuscle-F-actin: Tg(mylpfa:palm-mTurquoise2; mylpfa:LifeAct-mScarlet) images were acquired using Stellaris 8 with the following settings: (1) 440 nm WLL laser (CFP), 445- to 549-nm bandpass; (2) 569 nm (RFP), 600- to 755-nm bandpass; (3) pinhole size of 1 AU, line average of 1 or 4, and z-stack with 1-µm step size. Bright-field images were independently captured using a Leica M205 stereomicroscope (Leica LAS X) for measurements of standard length and body surface area.

## Image processing and analysis

Images were maximally projected and stitched using Image Composite Editor (Microsoft Research). For counting the myofibers and the nuclear numbers in each myotome, raw images were pre-processed with ImageJ (Fiji) by applying the Gaussian blur function (sigma value of 2). Then, the cross-sectional view of the myotome, including both dorsal and ventral compartments, was generated using the oblique section tool in Imaris (Bitplane 9.5). Total myofiber numbers were manually counted using the multi-point tool in ImageJ. Total nuclear numbers in each myotome were counted using the spot function in Imaris. In brief, individual myotome segments were outlined with the polygon selection tool. The following parameters were set for counting the nuclear number within each myotome: diameter, 7.0 µm; quality, 3.0; threshold for z-position adjusted for each image. For determining myotome volume, 3D surfaces were created using Imaris after pre-processing in ImageJ using the Gaussian blur (sigma value of 1). The dorsal and ventral sections of Myotome #12 were outlined with the polygon selection tool and trimmed. 2D anisotropic diffusion with "number of iterations 20" was applied to all trimmed images. A customized ImageJ script was applied for batch processing (see deposited code- Batch process myofiber volume). The myotome volume was determined with a surface thickness of 0.9 µm and a threshold value of 50. For determining individual myofiber volume, 3D surface rendering was performed with a surface thickness of 0.2 µm and threshold value of 240 in Imaris. For counting the nuclear number within each myofiber, images were processed using a customized ImageJ script (see deposited code- Nucleus within myofiber processing) to assign the centroid for each nucleus. The centroid was then added to the 3D images. The nuclear number was determined from the intensity sum of the centroids in each myofiber. For determining the tagged myofiber occupied area, stitched images were analyzed using the Otsu method. For counting the myofiber number in a cross-section, stitched images were segmented using Cellpose v2.2 (Waisman et al, 2021) with the following settings: model cyto and cyto2 with a cell diameter of 10–50 and a flow threshold of 0.6–1.0. The resulting ROI number was used as the tagged myofiber number. For counting the hue number in palmuscle-Multi, cross-sectional images were generated from each myotome. The delta-E (dE) value representing the distance between any two colors was set as 37.4 using a customized ImageJ script (see deposited code- Color analysis 1 and Color analysis 2). Fish drawings were created with BioRender.com.

## Histology and immunostaining

For cryosections, tissues were fixed in 4% paraformaldehyde (PFA) at 4 °C overnight. Fixed specimens were washed in PBS with 0.4% sucrose three times (5 min each). Tissues were then embedded in 1.5% agarose with 5% sucrose, followed by an incubation in 30% sucrose at 4 °C overnight. Frozen blocks were sectioned for antibody staining using a cryostat microtome at 16 µm (Leica CM3050s). The sections were washed in PBST three times (5 min each) and incubated in blocking buffer [NCS-PBST: 10% heat-inactivated newborn calf serum (NCS, Gibco, 26010066), 4% goat serum (Gibco, 16210072), and 0.1% DMSO in PBST] for 1 h at 37 °C. Samples were incubated with primary antibodies, F310 (fast myofiber; 1:100; Developmental Studies Hybridoma Bank,

AB_531863), F59 (slow myofiber; 1:50; Developmental Studies Hybridoma Bank, AB_528373), EGFP (1:250; GeneTex, GTX113617), or mCherry (1:250; GeneTex, GTX128508), in NCS-PBST for 3 h at 37 °C. Sections were washed three times in PBST (5 min each) and incubated with Alexa Fluor 594 anti-mouse (1:400, Invitrogen, A-11005) and Alexa Fluor 488 anti-rabbit (1:400, Invitrogen, A11034) secondary antibodies in NCS-PBST for 1 h at 37 °C. Nuclear DNA was stained with DAPI in PBST for 5 min before washing three times with PBST. Slides were mounted with Fluoromount G for imaging either with a Leica SP8 (HC PL APO 20x/0.75 CS2, WD 0.62 mm) or a Leica M205 stereomicroscope.

## Chemical treatments

For the induction of apoptosis, 2 dpf embryos were treated with cycloheximide (350 μM; Sigma-Aldrich, C1988) for 12 hr. For the induction of autophagic response, 4 dpf larvae were treated with rapamycin (1 μM; MedChemExpress, HY-10219) for 48 hr. For the blocking of autophagic responses, 14 dpf larvae were treated with chloroquine (50 μM; Sigma-Aldrich, C6628) for 48 h. For detecting the formation of the acidic autolysosomes, live larvae were incubated in LysoTracker Red DND-99 (10 μM; Invitrogen, L7528) for 1 h at 12, 13 and 14 dpf prior to imaging. For the overexpression of EGFP/ATG7, embryos at the one-cell-stage were injected with either *mylpfa:Tet3G-EGFP* or *mylpfa:Tet3G-EGFP-2A-ATG7* construct. The animals were then treated with fresh doxycycline (20 μg/ml; Sigma, D9891) on a daily basis starting at 9 dpf. For thyroxine (T4) hormone treatment, myofiber labeling in *Tg(palmuscle-Multi; myofiber:iCre#2)* was induced at 18 dpf. The animals were treated with T4 (MedChemExpress; HY-18341) under three different schemes: (1) T4 (30 nM and 100 nM) incubation (Brown, 1997) from 22 to 27 dpf; (2) T4 (30 nM) incubation from 22 to 27 dpf; (3) T4 (30 nM) incubation from 22 to 32 dpf. Animals treated with DMSO were used as a control.

## Single myofiber dissociation

Isolation of individual myofibers was performed as described (Horstick et al, 2013). Briefly, tissues were collected from *palmuscle-Dual* larvae (14 dpf), juveniles (70 dpf), and adults (1.7 years of age). Myofibers were dissociated for 1–2 h on a shaker (40 rpm) with DMEM (Gibco, 11995-065) and collagenase II (3 mg/ml; Worthington-biochem, LS004174). The dissociated samples were allowed to settle for 1 min, and the supernatant was removed. Again, fresh DMEM containing collagenase II was added to the tube before placing on a shaker for 2–4 h (35 rpm). The dissociated myofibers were then diluted with 1–2 ml DMEM and the suspension was transferred to a petri dish (Falcon; 353004). Myofibers were allowed to settle for 10–15 min prior to imaging (Leica SP8; 25x/0.95 HCXIRAPO).

## Bioenergetics assay of larval and adult myofibers

Larval and adult myofibers were freshly dissociated in bicarbonate-free DMEM medium (5% FBS, glutamine and penicillin-streptomycin). The dissociated myofibers were seeded in Seahorse XFp cell culture miniplates (Agilent, 103025-100) coated with Cell-Tak (Corning; 354240). For coating the wells, 15 μl Cell-Tak was mixed with 285 μl of 0.1 N sodium bicarbonate (pH 8.0) and added to each well. The plate was incubated at RT for 20 min, then washed with sterile water and air-dried prior to seeding of myofibers. Mitochondrial respiration was measured at the basal state using a Seahorse analyzer, followed by sequential injection of mitochondrial stress modulators (Agilent Seahorse XFp Cell Mito Stress Test Kit, 103010-100), oligomycin (2 μM), FCCP (0.625 μM), and actinomycin A/rotenone (0.5 μM). OCR (oxygen consumption rate) and ECAR (extracellular acidification rate) were measured using the Extracellular Flux Analyzer (Agilent, XF, Seahorse Bioscience). The basal OCR value was used for normalization. The spare respiratory capacity (SRC) was calculated as the difference between maximal and basal respiration. ECAR was calculated as the difference between maximal and basal ECAR values. Data were extracted by Wave software (version-2.6.3.5).

## CRISPR/Cas9-mediated knockdown of *atg7*

CRISPR sgRNA sequences were designed using the CHOPCHOP website (http://chopchop.cbu.uib.no/). The primer sequences for generating sgRNA templates, genotyping, and RT-qPCR are listed in Table EV1. sgRNAs targeting the gene *heatr6* (control) and *atg7* gene were synthesized using the MEGAshortscript T7 transcription kit (Invitrogen, AM1333) as described (Talbot and Amacher, 2014). sgRNAs targeting atg7 were injected (2 μl each, 2 μg/ μl) along with 2 μl Cas9 protein (1 μg/μl; PNA Bio, CP01-200) into zebrafish embryos at the one-cell stage. Pools of 30–50 injected larvae were used for genotyping and RT-qPCR analyses.

## RT-qPCR assays

Primers were designed to detect the transcripts from the recombined Brainbow cassettes, targeting the common sequences of mCherry, mYFP, and mCerulean. Whole animal samples were collected at 14, 28, and 42 dpf from pools of individuals ($n = 6–10$, 3, and 1 per group, respectively). At 70 dpf, animal tissues were collected from three different body compartments (i.e., anterior, middle, and posterior). Animals without Dox/Tam treatment were used as a control. For detecting *atg5, atg7, myhc4,* and *desma*, EK samples were collected from 14 dpf, 70 dpf, and 1.7-year-old fish. Juvenile/adult muscle tissues were collected from pools of 2–5 individuals. Skin and viscera were removed under a dissecting microscope prior to RNA extraction. Tissues were homogenized in 1 ml Trizol (Sigma-Aldrich, T9424) using TissueLyser II (Qiagen). RNA was extracted, and cDNA was synthesized from 2–3 μg RNA using the SuperScript III First-Strand Synthesis System (Invitrogen, 18080051). qPCR analysis was performed using a Roche Light-Cycler 480 according to the manufacturer's instructions. Each sample was analyzed in biological quadruplicate and technical triplicate as described (Wang et al, 2019). The primer sequences used for qPCR are listed in Table EV1.

## Statistical information

Statistical significance was analyzed with Prism 8.0 (GraphPad Software). A two-tailed Student's t test for parametric distributions was used when distributions passed the D'Agostino–Pearson normality test. A two-tailed Mann–Whitney test was used for non-parametric distributions. For analysis of three groups, a one-

way ANOVA test with Tukey correction for multiple comparisons was used when distributions passed the D'Agostino–Pearson normality test. Otherwise, a Kruskal–Wallis test with Dunnett's correction for multiple comparisons was used. Sample sizes and statistically significant differences are reported in figures.

## Data availability

The source data of Fig. 1G have been deposited at BioStudies, accessible with the following accession number: S-BSST1415. Customized scripts and usage instruction are available from Github: https://github.com/peggyscshu/Myotome-volume-Nucleus-count-and-Color-analysis).

The source data of this paper are collected in the following database record: biostudies:S-SCDT-10_1038-S44318-024-00136-y.

## Peer review information

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

## Acknowledgements

We thank Iris Chen and Taiwan Zebrafish Core Facility (TZCF) for zebrafish care and services. TZCF is supported by National Science and Technology Council (NSTC), Taiwan (NSTC 112-2740-B-400-001); Jennifer Liang, Stephan Schneider, Yi-Hsien Su, Yi-Ching Lee, Guang-Chao Chen, Chi-Kuang Yao, members of the Chen laboratory for comments on the manuscript; Marcus J. Calkins for English editing and comments; Keng-Hui Lin for Cell-Tak. We acknowledge funding support from Academia Sinica and NSTC to C.H.C. (AS-CDA-109-L03, AS-GCS-112-L01, NSTC 110-2628-B-001-016, and NSTC 111-2628-B-001-026).

## Author contributions

**Uday Kumar**: Conceptualization; Resources; Data curation; Software; Formal analysis; Validation; Investigation; Visualization; Methodology; Writing—review and editing. **Chun-Yi Fang**: Data curation; Formal analysis; Validation; Visualization; Methodology. **Hsiao-Yuh Roan**: Data curation; Formal analysis; Methodology. **Shao-Chun Hsu**: Software; Formal analysis; Visualization; Methodology. **Chung-Han Wang**: Resources. **Chen-Hui Chen**: Conceptualization; Resources; Data curation; Supervision; Funding acquisition; Investigation; Visualization; Methodology; Writing—original draft; Project administration; Writing—review and editing.

Source data underlying figure panels in this paper may have individual authorship assigned. Where available, figure panel/source data authorship is listed in the following database record: biostudies:S-SCDT-10_1038-S44318-024-00136-y.

## Disclosure and competing interests statement

The authors declare no competing interests.

# Expanded View Figures

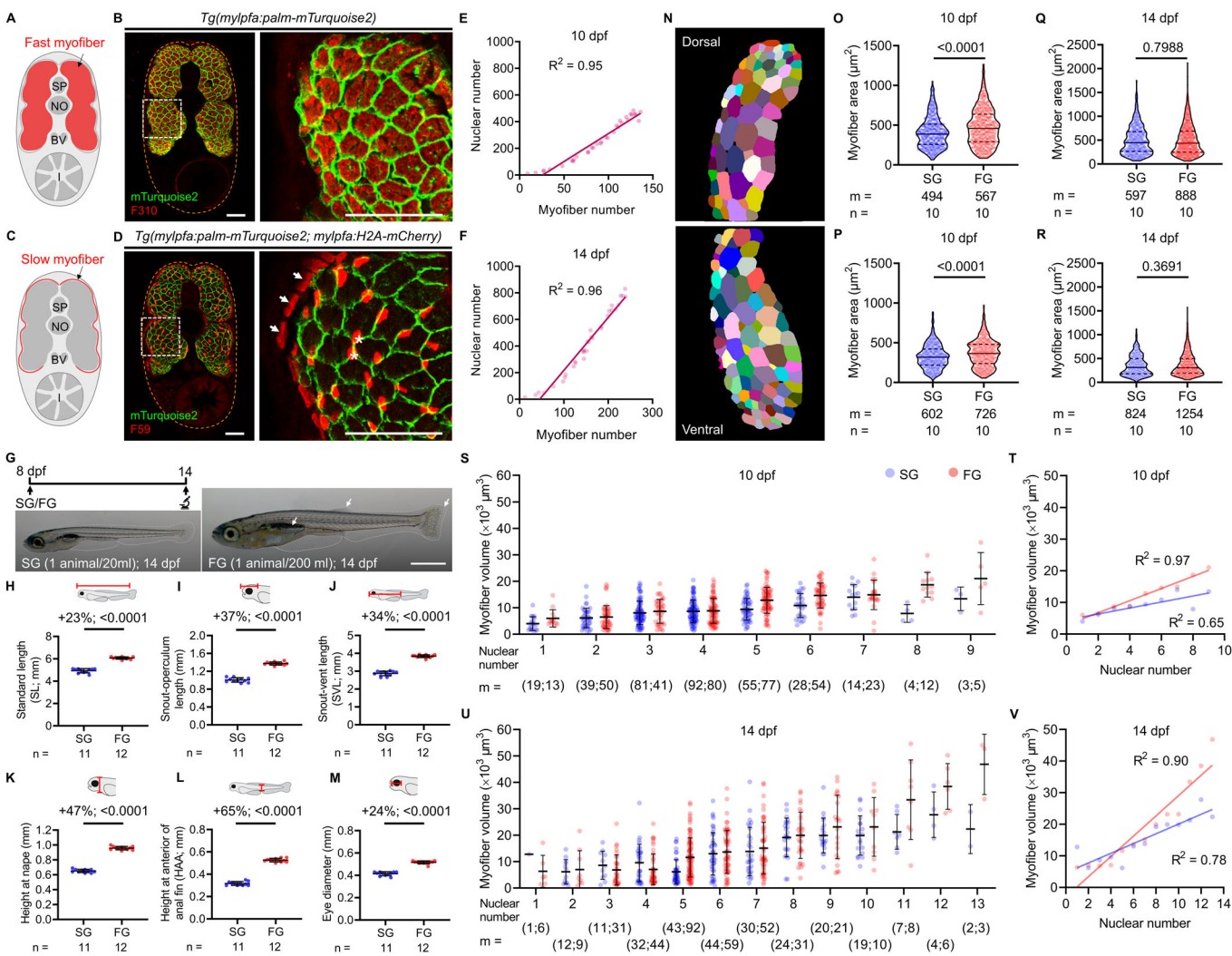

**Figure EV1. Myofiber volume scales linearly with myofiber nuclear number regardless of the growth conditions.**

(A) Schematic highlights the fast myofiber compartment. SP, spinal cord. NO, notochord. BV, blood vessels. I, intestine. (B) Histological examination of the labeled myofibers in the *Tg(mylpfa:palm-mTurquoise2)* transgenic line (left). White dashed box outlines the magnified region shown to the right. F310 Ab stains fast myofibers. (C) Schematic highlights the slow myofiber compartment. (D) Histological examination of the labeled myofibers in the *palmuscle-Dual*: *Tg(mylpfa:palm-mTurquoise2; mylpfa:H2A-mCherry)* double transgenic line (left). White dashed box marks the magnified region shown to the right. Orange dashed lines outline the entire trunk compartment. F59 Ab stains slow myofibers. White arrows point to slow myofibers. White asterisks mark the fast myofiber nuclei. (E, F) Linear regression coefficients analyses of myofiber number versus nuclear number across all 32 myotomes in individual animals at 10 dpf (E) and 14 dpf (F). (G) Timeline of the growth manipulation and tracking scheme. Representative images of fish grown under either slow growth (SG; 1 animal in 20 ml) or fast growth (FG; 1 animal in 200 ml) conditions at 14 dpf. White arrows point to the emergence of swim bladder, median fin fold, and caudal fin structures under the FG condition. White dashed lines mark the fin fold region. (H–M) Quantification of anatomical traits under SG and FG: Standard length, SL (H); Snout-operculum length (I); Snout-vent length (J); Height at nape (K); Height at anterior of anal fin, HAA (L) and Eye diameter (M). (N) A cross-sectional image of a myotome, individual myofibers are depicted in pseudocolor by Cellpose. (O–R) Quantitation of the myofiber cross-sectional area in either the dorsal or ventral compartment of Myotome #12 at 10 dpf (O, P) and 14 dpf (Q, R). (S) Quantification of individual myofiber volume at 10 dpf under either slow growth (SG) or fast growth (FG) conditions. (T) Linear regression coefficients were calculated for myofiber volume versus nuclear number (10 dpf). (U) Quantification of the individual myofiber volume at 14 dpf under either the SG or the FG conditions. (V) Linear regression coefficients were calculated for myofiber volume versus nuclear number (14 dpf). Data from biological replicates are shown as mean ± standard deviation (H–M, S, U) or violin plots (solid lines, median; dashed lines, quartiles; O–R). Significance was examined by examined by two-tailed Student's t-test (H–L) or two-tailed Mann–Whitney test (M, O–R). *P* values are shown above the horizontal lines for intergroup comparisons. *n* = number of animals (H–M, O–R) and m = number of myofibers (O–S, U). Four animals (E, F) and ten animals (S, U) were used in each condition. Scale bars, 50 μm (B, D); 1 mm (G).

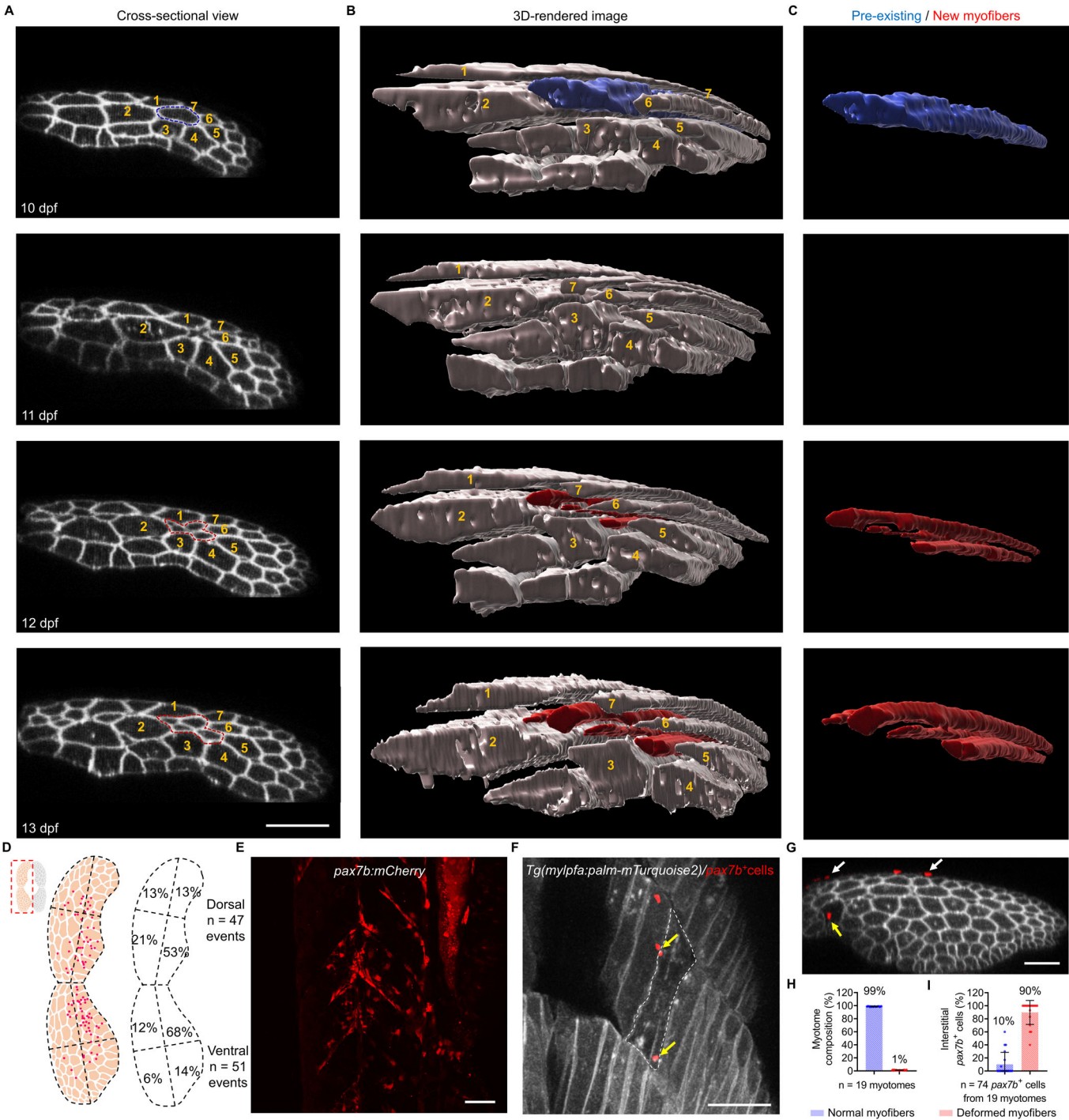

**Figure EV2. Individual deformed myofibers are fully eliminated and replaced.**

(A–C) Time-lapse images of the same myotome over a 3-day period, showing the myofiber elimination and the replacement processes in a cross-sectional view (A) or as a 3D rendering (B, C). Blue dashed lines outline a deformed myofiber. Red dashed lines outline newborn myofibers. Neighboring myofibers are labeled with respective numbers. (D) Spatial distribution of deformed myofibers in the dorsal and ventral myotomes. A total of 98 dissolution events were captured and mapped. (E) Representative image of *pax7b*-positive muscle stem cells expressing mCherry. (F) Representative image showing *pax7b*-positive cells (yellow arrows) in close proximity to a deformed myofiber. (G) Cross-sectional view of a myotome showing *pax7b*-positive cells located either at the myotome's periphery (white arrows) or within the interstitial space (yellow arrows). (H, I) Quantification of myofiber composition within a myotome (H) and the interstitial *pax7b*-positive cells closely associated with either normal or deformed myofibers (I). Data from biological replicates are shown as mean ± standard deviation (H, I). *n* = number of myotome (H) or cells (I). Scale bar, 50 µm (A, E–G).

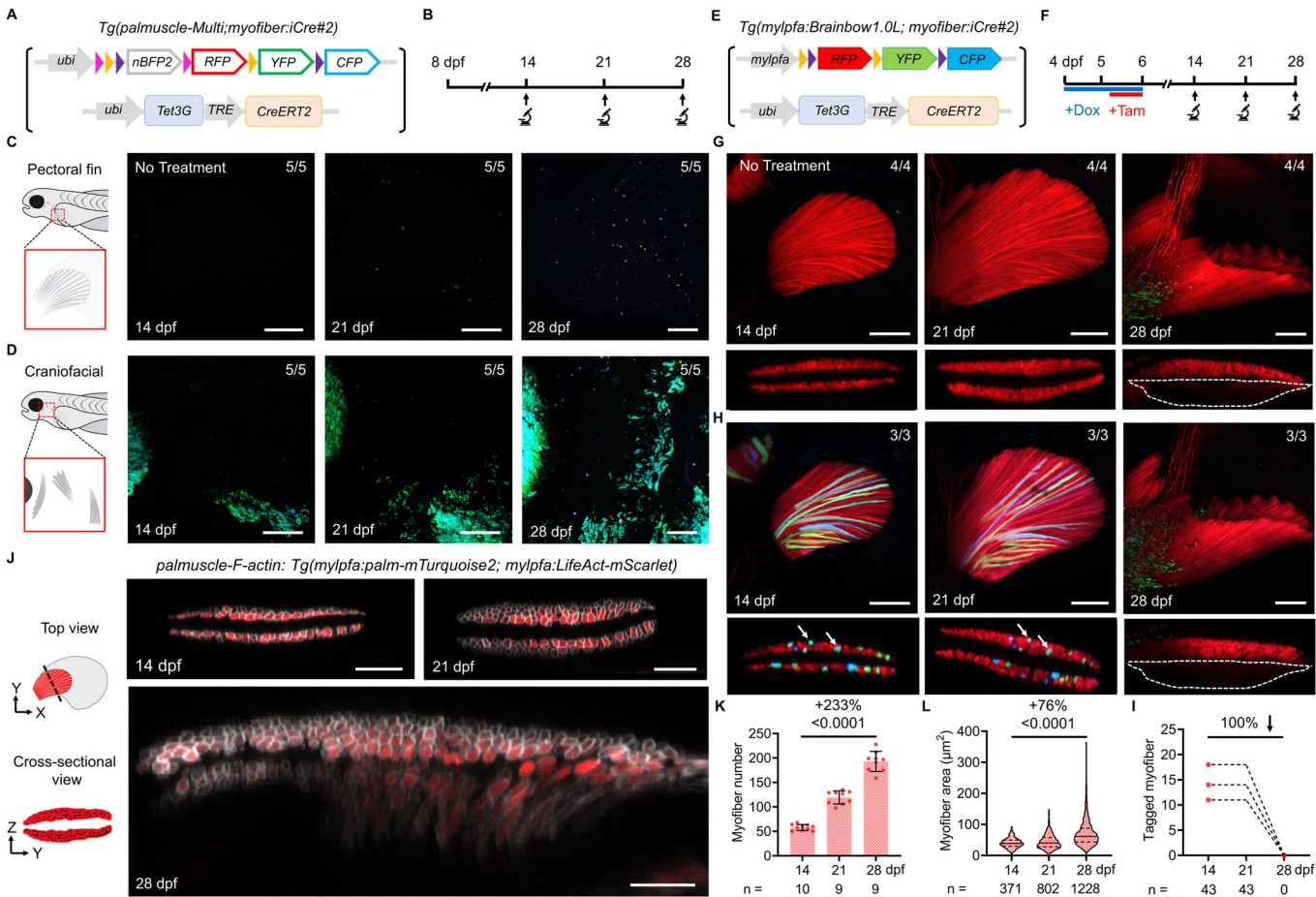

**Figure EV3. Pre-existing pectoral fin myofibers are eliminated and replaced by a de novo source.**

(A) The *palmuscle-Multi* and the *myofiber:iCre#2* transgenic constructs. (B) Timeline of the tracking scheme. (C, D) Long-term tracking of the *Tg(palmuscle-Multi; myofiber:iCre#2)* double transgenic line in different anatomical regions, including the pectoral fin region (C) and the craniofacial region (D). No leaky Cre activity was observed. (E) The *mylpfa:Brainbow1.0 L* and *myofiber:iCre#2* transgenic constructs. (F) Timeline of the treatment and tracking scheme. (G) Long-term tracking of the *Tg(mylpfa:Brainbow1.0 L; myofiber:iCre#2)* double transgenic line showed no leaky Cre activity. (H) Long-term time-lapse imaging of the same myofibers at 14, 21, and 28 dpf. White arrows highlight "disappearing myofibers". White dashed lines (bottom right, G and H) outline the bottom pectoral fin myofiber compartment, which becomes less visible at 28 dpf due to tissue thickening. (I) Quantification of tagged myofiber numbers. (J) Cross-sectional view of the pectoral fin myofibers showing both top and bottom layers at 14, 21 and 28 dpf. (K, L) Quantification of myofiber numbers (K) and myofiber areas (L). The entire top layer of fin myofibers were included in the quantification. Data from biological replicates are shown as mean ± standard deviation (K) or violin plots (solid lines, median; dashed lines, quartiles; L). Significance was examined by two-tailed Student's t-test (K) or two-tailed Mann–Whitney test (L). Percent differences and $P$ values are shown above the horizontal lines for intergroup comparisons. $n$ = number of animals (C, D, G, H, K) or myofibers (I, L). Scale bars, 50 μm (J) and 100 μm (C, D, G, H). dpf, days post-fertilization.

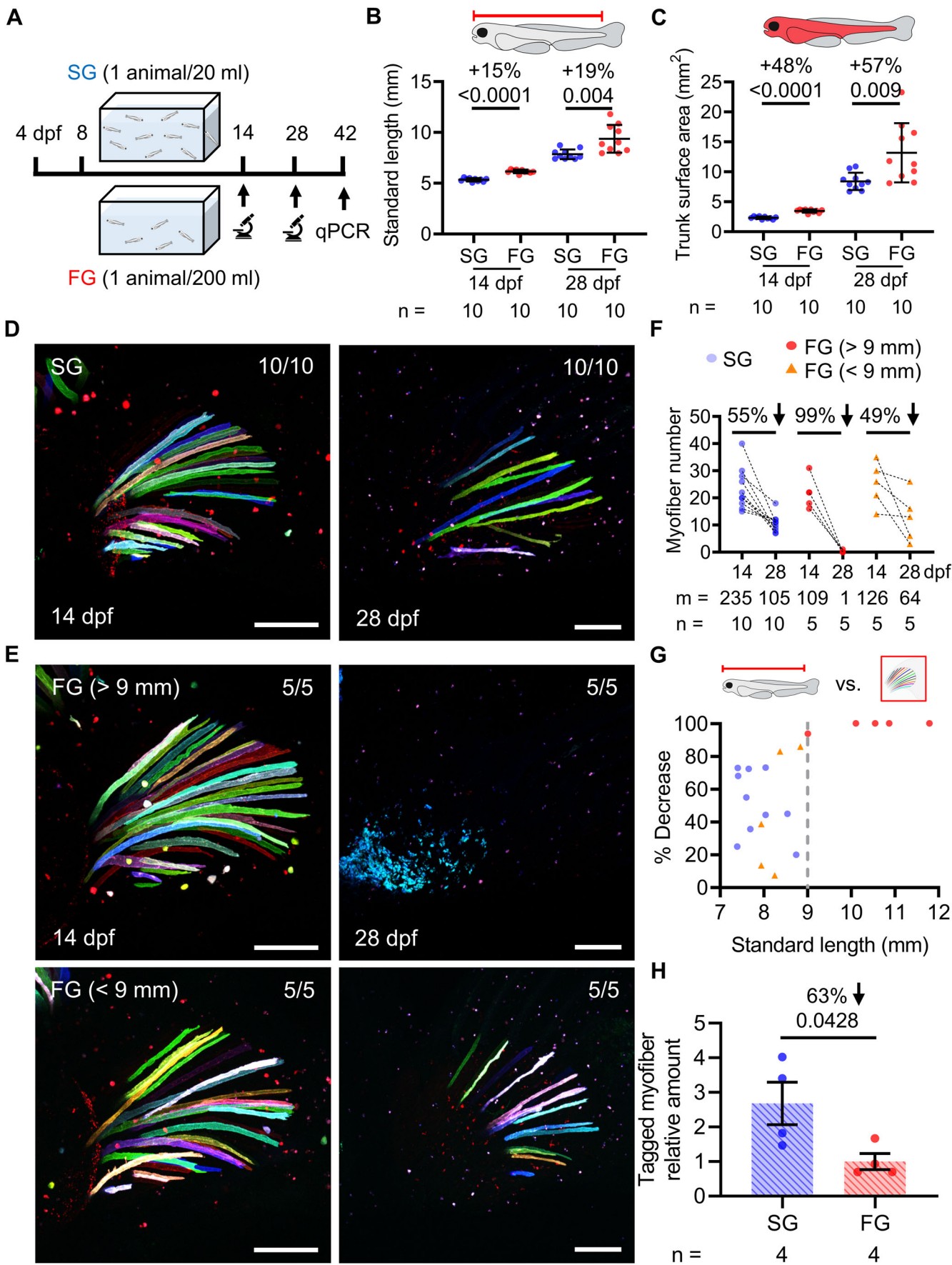

◀ **Figure EV4. Rapid growth condition promotes myofiber elimination.**

(A) Timeline of larval growth manipulation and tracking scheme. SG, slow growth. FG, fast growth. (B, C) Larval growth under the SG and FG conditions as determined by standard length (B), and trunk surface area (C). (D, E) Long-term tracking of the *Tg(palmuscle-Multi; myofiber:iCre#2)* double transgenic line under SG (D) and FG (E) conditions. (F, G) Quantification of tagged myofiber numbers (F) and decreases in percentage (G). Gray dashed line highlights the standard length of 9 mm. (H) RT-qPCR analysis of whole-animal myofiber loss at 42 dpf under either SG or FG conditions. Data from biological replicates are shown as mean ± standard deviation (B, C) and mean ± standard error (H). Significance was examined by two-tailed Student's t-test. Percent differences and *P* values are shown above the horizontal lines for intergroup comparisons. n = number of animals (B–F) or biological replicates (H). m = number of myofibers (F). Scale bar, 100 μm (D, E). dpf, days post-fertilization.

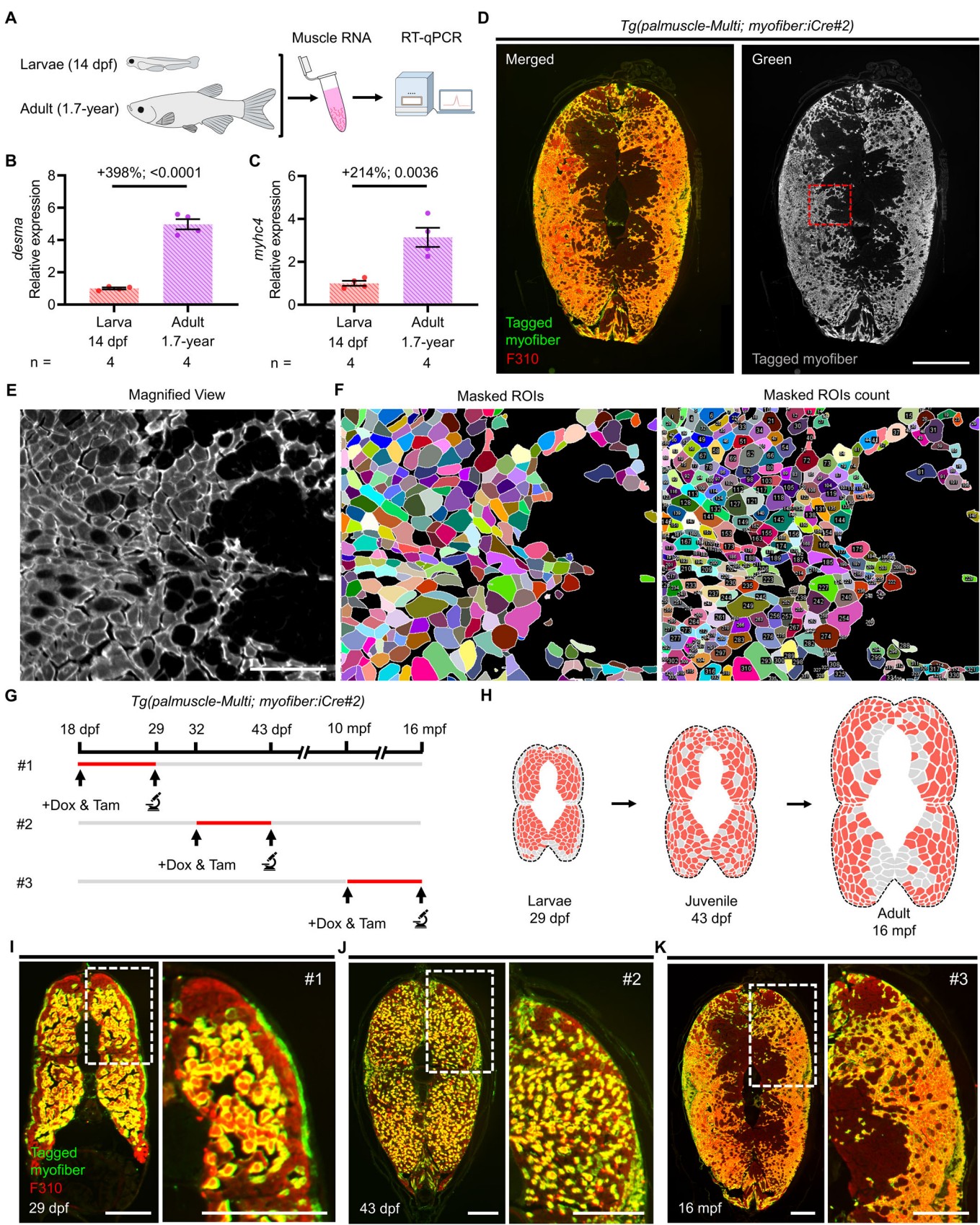

**Figure EV5. Larval myofibers are functionally distinct from adult myofibers.**

(A) Illustration of zebrafish at larval and adult stages, reflecting their relative size differences. (B, C) RT-qPCR analyses of expression of the intermediate filament *desma* (B), and sarcomere myosin *myhc4* (C) in whole animals (14 dpf) or dissected muscle tissues (1.7 years of age). (D) Cross-sectional image of the middle-trunk region of the animal at 16 mpf with tagged myofibers. F310 Ab stains fast myofibers. (E) Magnified view of the trunk region indicated in (D) by red dashed box. (F) Individual tagged myofibers from cross-sectional images shown in pseudocolor (left) and with respective ROI numbers (right). (G) Timeline of the treatment and tracking scheme. (H) Schematic drawing of the progressive shift in hyperplastic growth zone across different developmental stages. (I–K) Histological examinations of the tagged myofibers in the middle-trunk region from larvae to adult stage, showing growth zone shift from the periphery at 29 dpf (I), to the interstitial space at 43 dpf (J), and ultimately to the deep region of the myotome at 16 mpf (K). White dashed boxes outline the magnified regions shown to the right of each panel. F310 Ab stains fast myofibers. Data from biological replicates are shown as mean ± standard error (B, C). Significance was examined by two-tailed Student's t-test. Percent differences and *P* values are shown above the horizontal lines for intergroup comparisons. *n* = the number of biological replicates. Scale bars, 1 mm (D); 200 μm (E, I); 300 μm (J); 500 μm (K). dpf, days post-fertilization; mpf, months post-fertilization.

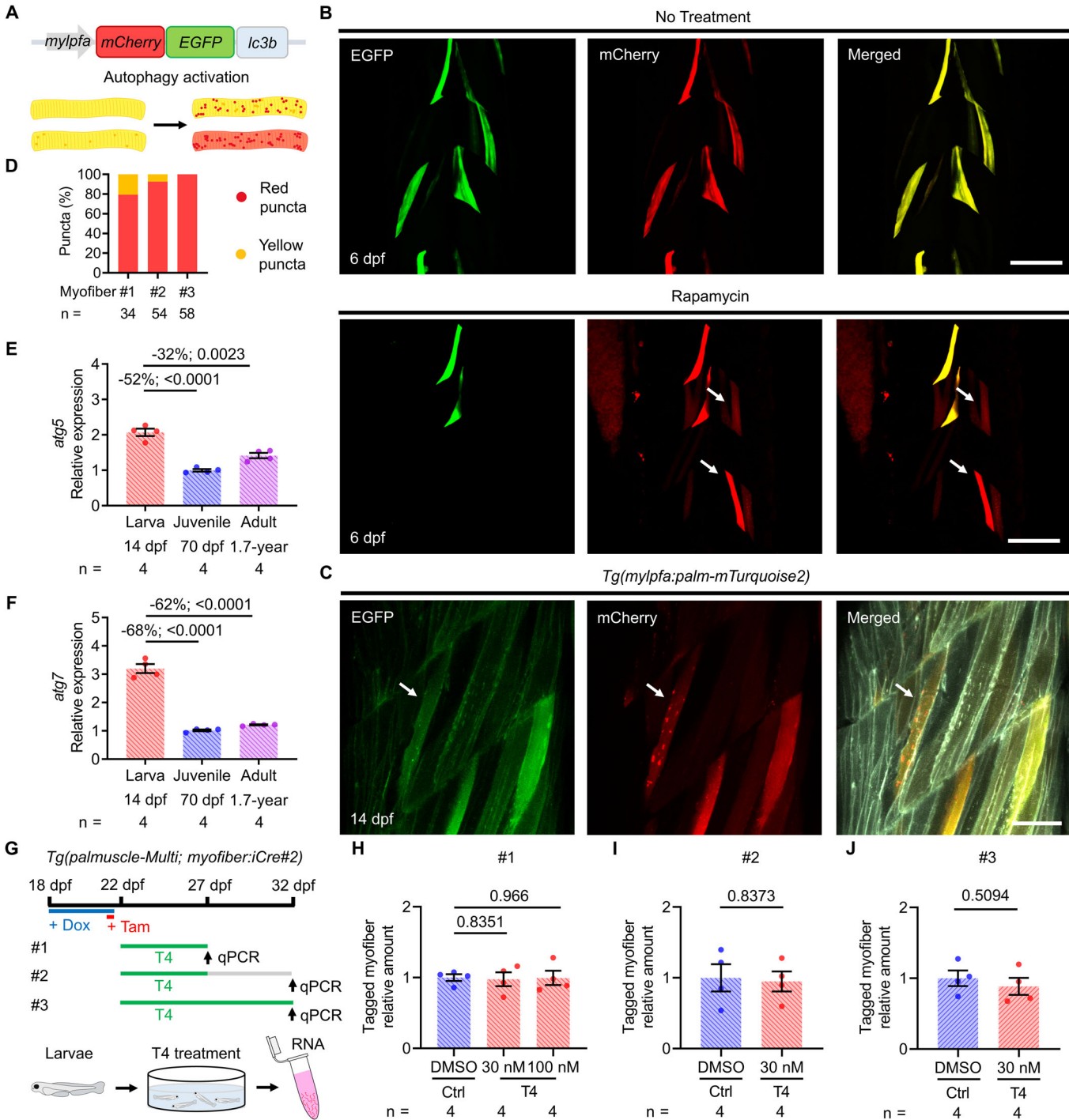

**Figure EV6. Live monitoring of autophagic activation in deformed myofibers.**

(**A**) The transgenic construct for live monitoring of autophagic activation in myofibers. (**B**) Treatment with rapamycin activates autophagic response in myofibers. White arrows point to activated, mCherry-positive myofibers. Of note, without the treatment of rapamycin, all 6 dpf myofibers were both EGFP- and mCherry-positive. (**C**) Deformed myofibers contain abundant red puncta. White arrow points to a deformed myofiber. (**D**) Percentages of red and yellow puncta in three individual deformed myofibers. (**E**, **F**) RT-qPCR was performed to analyze expression of the autophagic genes *atg5* (**E**) and *atg7* (**F**) in whole animals (14 dpf) or dissociated myofibers (70 dpf and 1.7 years of age). (**G**) Timeline and three different thyroxine hormone (T4) treatment schemes. (**H–J**) RT-qPCR analysis of the whole-animal myofiber loss for each of the schemes. Data from biological replicates are shown as mean ± standard error (**E**, **F**, **H**, **I**, **J**). Significance was examined by two-tailed Student's t-test. Percent differences and *P* values are shown above the horizontal lines for intergroup comparisons. *n* = number of puncta (**D**) or biological replicates (**E**, **F**, **H**, **I**, **J**). Scale bars, 100 μm (**B**); 50 μm (**C**). dpf, days post-fertilization.

