## [Peer Review File · The EMBO Journal]

Whole-body replacement of larval myofibers generates permanent adult myofibers in zebrafish

Chen-Hui Chen, Uday Kumar, Chun-Yi Fang, Hsiao-Yuh Roan, Shao-Chun Hsu, and Chung-Han Wang

Corresponding author: Chen-Hui Chen (chcchen@gate.sinica.edu.tw)

Review Timeline:

Submission Date:	7th Mar 24
Editorial Decision:	24th Apr 24
Revision Received:	6th May 24
Accepted:	17th May 24

Editor: Ieva Gailite

Transaction Report:

(Note: The initial review process for this manuscript took place with another journal. The initial reviewers' comments and authors' responses for this article have been made available. With the exception of the correction of typographical or spelling errors that could be a source of ambiguity, letters and reports are not edited. Depending on transfer agreements, referee reports obtained elsewhere may or may not be included in this compilation. Referee reports are anonymous unless the Referee chooses to sign their reports.)

Referee #1:

Overall comments:

This gorgeous manuscript really puts the zebrafish through its paces, giving a critical insight into muscle biology. Their cornerstone conclusion (that myofiber elimination could be a means to promote muscle growth) is both completely new and counter-intuitive. It differs both from how I think of muscle growth, and growth generally. Such a claim needs strong support, and indeed the authors provide excellent experimental support for this cornerstone conclusion.

The in toto lifetime imaging of zebrafish through their first month of life is both unexpected and powerful - especially when paired with whole myofiber sampling through later phases of life. This study also provides a comprehensive atlas of muscle growth in zebrafish, and conducts dozens of impactful experiments that shift how I understand the timing, location, and mechanism of muscle growth. All this while using compelling data at every step and fitting nicely with established knowledge. The authors tunnel deeply into each of their questions, and rigorously validate their central claims. The logic of this paper is sound and I do not feel compelled to request any new experiments. There are a few points where I push back on author interpretations, placed towards the top of the minor comments section, and the remainder are mostly issues of presentation.

In summary, I hope that this work is swiftly given press at the highest tier.

Response (1.1): We are grateful for Reviewer #1's enthusiastic comments, and have addressed each of the points raised, as outlined below.

Minor comments:

Figure 6F is not compelling and could be removed or reinterpreted without detriment to the paper. This panel claims to show an increase of sarcomere length with age; however, an uptick of only 1% is seen between juvenile and adult and that tiny shift appears to reverse at 1.7 years. The authors reach this concluding using an ANOVA post-hoc that may be misleading because the Juvenile and Adult distributions are bimodal, though Tukey's comparison is a better choice in this case than Krustal-Wallis which shouldn't be used to determine a mean. Also, the sarcomere length determination is not mentioned in methods. By contrast, the other data shown at the top of figure 6 (B-E) is compelling, and makes the authors central point (just not the added point about sarcomere lengths). This one panel and its conclusions stand in contrast with statistical analysis that is excellent in the paper overall.

Response (1.2): We thank Reviewer #1 for their feedback on the figure and statistics. Following the suggestion, we have now removed Figure 6F.

Line 120: The authors claim that it is intriguing that somite number is constant in fish as they grow to adulthood. This finding is not surprising or new - recall that each somite corresponds to one offset vertebrae; the number of vertebrae in zebrafish is firmly established. There are plenty of references

showing that somite count peaks fairly early in zebrafish and corresponds to vertebral formation. cfi Morin-Kensicki 2002. Morrow 2017 shows that counts attain their plateau of 33 somites by 36 hpf. Plenty of other resources are available. Or perhaps the intrigue was that myofiber counts and size increase - which is also established, cfi Patterson 2008 or Roy 2017.

Response (1.3): We thank the reviewer for the careful reading, and have updated the references and revised the text on page 6, paragraph 1:

“In line with research on early zebrafish development (10-12), the total myotome number during the post-embryonic growth phase remained largely constant (34 per animal on average; $n = 4$; Fig. 1F); the total myofiber number had a sharp (157%) increase by 14 dpf (from 1942 ± 148 to 4988 ± 497 per animal; $n = 4$; Fig. 1G).”

Line 150: Figure 2I shows that myofiber volume increases rapidly to the 14 day size when grown at low density, it is inaccurate to say that it plateaus at 14 dpf, especially since in Figure 4D the authors very clearly show that myofiber is larger in adults. I suspect they say "plateau" to implying this is a cap for the early-larval fibers, which later die... but the overall phrasing in this sentence is not working for me. Likewise, the claim on line 480 that larval myofiber size is limited (based on this finding in line 150). This conclusion seems like even more of a stretch, given that the authors did not test the possibility of growth past this timepoint. To make this claim, the authors would need to block fiber death and raise past 14 dpf, but they could very easily fix the problem with rephrasing.

Response (1.4): As advised, we have revised the text for improved clarity on page 7, paragraph 1 and page 21, paragraph 2:

“Despite the ongoing increase in the overall myotome volume, the growth of each myofiber appeared to reach a temporal constraint at 14 dpf regardless of the rearing conditions (Fig. 2 G-I) and anatomical locations (SI Appendix, Fig. S1 N-R).”

“Consistent with this notion, we determined that (1) the size growth of larval myofibers is temporally constrained (Fig. 2K);”

Line 162: It is true that a consistent myonuclear domain size was shown in mouse in 2020 for a given stage/fiber type (Ref 9). That same year, a study in zebrafish reached the same conclusion (Ref 5). As such, it would be appropriate to alter the sentence in a way that includes this zebrafish study. Mention of this prior work in no way diminishes the current one - especially given how thoroughly the current authors test the concept and show that growth conditions can alter the ratio of cytoplasm to nuclear count.

Response (1.5): We are grateful for the comment and have revised the text on page 8, paragraph 2:

“Consistent with studies in both mice and zebrafish that showed nuclear numbers may themselves promote hypertrophic growth in myofibers (5, 9), we found that the nuclear number within a myofiber exhibited a robust, linear correlation with the myofiber volume,”

It's good to see new myogenic labels to augment existing resources like Musclebow, Musclebow2, and 3MuscleGlow. The titles are clever- however, the superscript in "Palmmuscle^{dual} " and "Palmmuscle^{multi} " is not good nomenclature. This superscript implies that "Dual" and "Multi" are different alleles of the same gene, which isn't true. Instead, the designators could be given a hyphen with no superscript: "Palmmuscle-Dual" and "Palmmuscle-Multi". Optional: If the authors are going to give clever brand names to transgene combinations, they might also consider naming the F-Actin/Membrane combination, which seems like it will be at least as useful as the membrane/nuclei combination.

Response (1.6): We appreciate the reviewer for pointing this up, and we have accordingly updated the names of those three tools. Now read as: "*palmmuscle-Dual*", "*palmmuscle-Multi*", and "*palmmuscle-F-actin*".

Presentation issues:

The Palmmuscle label becomes even messier in the annotation on Figure 1a, "palmmuscle^{dual}; Tg(mylpfa:palmm-turquoise2; mylpfa:H2A-mCherry)" because the first semicolon implies that palmmuscle^{dual} is some third transgene, and the pairing of two transgenes within one Tg paranthesis implies that this is a single tandem transgene, instead of two segregating alleles as reported in the methods. Proper nomenclature seems to be "palmmuscle^{dual}: Tg(mylpfa:palmm-turquoise2)^{as69}; Tg(mylpfa:H2A-mCherry)^{as70}". Note that the transgene alleles are written here without superscript, following ZFIN conventions.

Response (1.7): We very much appreciate the suggestion, and have made the respective correction. The annotation on Figure 1A now read as: "*palmmuscle-Dual: Tg(mylpfa:palmm-turquoise2); Tg(mylpfa:H2A-mCherry)*".

The authors provide helpful overview illustrations to describe how they took measurements in Figure 1D-F and S1H-M. However, it's a little confusing because the head in these illustrations is shaped like a 48 hpf zebrafish (no jaw) instead of an 8-14 dpf zebrafish (snout/jaw protrudes in front of the eye). These little illustrations could be adjusted for clarity.

Response (1.8): We have now corrected the zebrafish larvae illustrations throughout on Figures 1D-F, 2C-E and 2G, 4I, 5B-D and 5G, 6A, 7H and 7X, S1H-M, S3C and S3D, S4B, S4C and S4G, S5A, and S6G to accurately reflect their appearance at the larval stage.

Line 110: The phrase "fast myofibers (the skeletal myofiber equivalent in zebrafish)." needs adjustment, because zebrafish of course have slow myofibers, and by adulthood intermediate fiber types are present too. Perhaps this could read along the lines of "fast myofibers (the predominant skeletal myofiber type in zebrafish)".

Response (1.9): We have revised the text on page 5, paragraph 2 as advised:

"fast myofibers (the predominant skeletal myofiber type in zebrafish)"

Some figures zig-zag and spiral in a non-intuitive fashion. For instance, Figure 2 zig-zags A-E left right then F-G down then H up, etc. Likewise, the pattern of Figure 7 panels progress like a confused nautilus. These should be rearranged into a more linear order.

Response (1.10): We have now rearranged both figures for improved readability.

Figure 3 contains lovely overviews, zooms, and explanatory illustrations. It is almost easy to interpret - but with this perfect gridwork of panels, my eye has a hard time focusing on any one time-lapse series. I suggest adding differently colored background shading behind panels (e.g. gray shading behind G, brown behind H) to help the eye lock into individual time-lapse compilations.

Response (1.11): Excellent suggestion. We have now added distinctively colored boundaries around the individual panels to Figures 3G, 3H, 3I, 3J, and 3K, aligning them with their corresponding quantifications depicted in Figure 3M.

Likewise, the paper would benefit from addition of at least one supplemental video showing the replacement process. I am not requesting new experiments - simply a resource that puts existing frames together in video format so readers can scroll back and forth through time-points. This would also enable the authors to show intermediate imaging days that are truncated in panels G-K.

Response (1.12): As advised, we have added one additional movie, Movie S5, showing the entire replacement process.

The authors state that myofibers are usually lost in the deep-center of myotomes, but the first example sits on the medial edge (Figure 3F) - and elsewhere they show that the whole original myotome is lost. To me, deep-center reads as "middle of the muscle bulk" (e.g. Figure S2A) rather than medial edge. This read is also complicated by later data showing that new fibers are eventually layered into whole medial third of the myotome. Could the first example be chosen to reflect the typical positioning - or should the locational description be adjusted?

Response (1.13): We have revised the sentence on page 9, paragraph 1 to more accurately describe the location of the myofiber loss.

"Notably, these dissolution events appeared to exhibit a spatial bias favoring the medial edge compartments"

Do the authors ever catch the Pax7b cells during mitosis in the replacement zone? Such events may be rare given that they are "only" sampling 2x/day, but supplemental examples that mitosis has been observed could strengthen this already strong story.

Response (1.13): As the reviewer pointed out, our imaging interval is not sufficient to catch the Pax7b cells that are dividing in the replacement zone. Given that capturing the rapid process will require additional imaging tools, we kindly propose to characterize the dynamics of Pax7b cells and their interaction with degrading myofibers in a subsequent study.

It's tough to see the sparse myofibers in Figure 5H,L,P because there is so little tag-signal and so much F310. The F310 channel could be dimmed to help those rare myofibers stand out.

Response (1.14): Once again, we thank the reviewer for the great suggestion. We have adjusted the F310 channel to enhance the visualization of those tagged myofibers in Figures 5H and 5L.

Readers will dream of a subsequent methods paper showing a version of Cre that labels all fast-twitch myofibers without any leakiness - I hope that the authors will work on building one, but it is outside the scope of the present work.

Response (1.15): We very much appreciate the reviewer's encouragement and recognition of our years-long efforts to identify a transgenic Cre line devoid of leakiness. This is crucial for ensuring the accuracy of long-term cell tracking studies, a consideration that is often overlooked. In fact, we are still screening for Cre lines that could label both fast and slow myofibers in full without leakiness. We hope to pursue a subsequent study on this muscle swapping phenomenon.

Referee #2:

Comments:

This paper contains data interpreted to support a remarkable conclusion, that most or all larval myofibers are replaced entirely during the late larval/early adult period. Such extraordinary claims require extraordinary evidence to justify space in premier/career-building journals. It is necessary for reviewers to be rigorous in such situations.

I already reviewed this manuscript at another journal. The data is essentially unchanged from the original version (just some panels moved around) and the text conclusions and interpretation is almost identical. Therefore, my conclusions and arguments from the previous round of review are still valid.

Response (2.1): We have revised the manuscript by incorporating additional figures to address each of the raised comments, as outlined in the respective responses provided below.

The evidence for death and replacement of myofibers at some rate is fairly convincing from the timelapse images shown. It is hard, of course, to eliminate the possibility that this reflects light/heat/free radical-induced toxicity from the initial confocal scans. So the rate of such events in non-imaged individuals is uncertain. I think such events have been reported before, for example in the sibling controls of dystrophic mutant fish, which mutants undergo extensive fiber detachment and death.

Response (2.2): In the study, we provide three independent lines of evidence supporting the conclusion of extensive myofiber death (Figures 5 and S4): (1) live tracking of individual myofibers throughout the post-embryonic phase, (2) histological examinations of the tagged myofibers, and (3)

in toto qPCR-based detection of the transcripts resulting from the recombined tagging cassettes. As noted by the reviewer, excitation light-induced toxicity could be a concern in any live imaging-based studies. We would like to emphasize that the methods (2) and (3) do not rely on confocal scans of live animals. Yet, all the results collectively support the main conclusion of our study, which is that the majority of larval myofibers are entirely eliminated as an individual reaches adulthood.

Response (2.3): Since the reviewer did not provide a specific reference, we thoroughly reviewed several studies investigating zebrafish dystrophic mutants. As noted by the reviewer, these studies nicely demonstrate the connection between myofiber detachment and subsequent cell death in the mutants (Bassett et al., 2003; Hall et al., 2007; Jacoby, et al., 2009). However, none of these studies provided results or observations regarding myofiber death in wild-type control siblings. Moreover, it is important to note that all the aforementioned mutant studies focused on the development of early embryonic myofibers, whereas our study investigates the growth of post-embryonic myofibers.

In my view the issue of widespread fiber replacement is not yet sufficiently strongly supported.

I see three major problems:

1. How do the authors know that the loss of detectable label in marked fibers is not simply dilution of the label in the increasingly-large labelled adult fibers? Note that in Fig 5H the scale bars are different sizes in 14, 42 and 70 dpf images reflecting the massive growth in 3 dimensions, which even for a cylinder goes up as the square of the linear dimension i.e. about 100-fold between 14 and 70 dpf, paralleling the increase in fiber cross sectional area reported by Roy et al (2017). This dilution would a) prevent detection in the sectioned material, b) make live scanning deep in the tissue of a larger larval fish difficult due both to surface pigmentation and the depth of tissue causing light refraction, c) cause dilution of any expressed mRNA such that the observed QRT-PCR dilution of the mRNA derived from a few labelled nuclei is what would be expected anyway.

Response (2.4): While we acknowledge the reviewer's concern regarding the significant growth observed during these time points, we wish to emphasize that our conclusion of extensive myofiber death is supported by three independent approaches: (1) continuous live tracking of approximately 500 individual myofibers across two different transgenic lines (see Figures 5A-5F and Figures S3E-S3I), (2) millimeter-scaled histological examination of the tagged myofibers at single-cell resolution (see Figures 5G-5N and Figures S5D-S5F), and (3) whole-body qPCR-based detection of the recombined cassettes (Figures 5R-5T'). Since each of the three approaches has its respective strengths and limitations, we have revised the text on page 13, paragraph 2, to provide clarification:

"Intriguingly, yet in line with the histological examination, we detected near-background readouts from all three body compartments (an average of 99% reduction; Fig. 5 T and T'). Despite the method being unavoidably influenced by the dilution effect from natural animal growth, the findings affirm the near-complete absence of the recombined cassettes on a whole-animal scale. By synthesizing evidence from live tracking of individual myofibers, histological examination, and *in toto* detection of the recombined cassettes, we conclude that the majority of larval

myofibers exist at negligible levels as an individual reaches adulthood, irrespective of anatomical position.”

2. Shutdown of the recombined transgenes as the animals mature has not been eliminated (for example, explaining the dramatic pectoral fin change in only 2 weeks in Fig 5B). While the authors could use PCR detection of loss of the deleted/recombined genomic DNA, this would also require exceptionally rigorous quantitative analysis to eliminate the possibility of dilution due to the total increase in unrecombined DNA within the muscle tissue as the authors show that many nuclei are added during later growth.

Response (2.5): We have now included data demonstrating that the reporter/recombined transgene is not silenced across various developmental stages (See Response 2.7).

Response (2.6): In response to the reviewer's concern regarding the potential dilution effect caused by animal growth, we have revised the text to explicitly state the limitation of qPCR-based assays, along with emphasizing the collective evidence presented in the study for supporting our main conclusion (see Response 2.4).

3. Given the reported ubiquitous expression of the ubi: promoter, the selection of a line only showing expression in fast muscle fibers suggests that the transgene in this line may have integrated in a locus that is open in fast muscle but suppressed everywhere else. How do the authors know, it is not only open in young fast muscle, shuts down older (unless re-opened by recombination, as in the 10 mpf experiment)?

Some of these issues could be eliminated by driving recombination later (but not as late as 10 months when fibers are already large) and at low frequency (say at 42 dpf yielding <10 labelled fibers per myotome a few days later) and then showing the marked fibers remain present and detectable in cross sections at a similarly low frequency at 180 dpf. However, it would be difficult to eliminate the possibility that the induction of enough recombination to label the large 42 dpf fibers did not reflect more total recombined nuclei than in the 4-6 dpf Tx treatment, and thus explain any perdurance to 180 dpf. The data in Fig 6K suggests such perdurance, but the level of recombination is high (the high intensity of label and rather uniform colour pattern confirms that many nuclei in each fiber had recombined).

Response (2.7): To exclude the potential silencing of the reporter transgene in older fish, which could account for the loss of myofiber signal, we had examined the reporter activity at three distinct developmental stages. We found no evidence of developmental stage-specific silencing effects during both post-embryonic growth and adulthood. We have now incorporated the results as Figures 5O, 5P and 5Q, and updated the text on page 12, paragraph 1:

“To preclude the possibility of potential silencing of the reporter *palmsmuscle-Multi* at later stages, which could result in the loss of myofiber signals, we examined reporter expression by inducing Cre activation at three distinct time points (i.e., 18 dpf, 32 dpf, and 10 mpf; Fig. 5O). The results consistently demonstrated robust labeling of myofibers upon Cre activation, assuring that the

reporter maintains steady expression levels throughout larval, juvenile, and adult stages (Fig. 5 P and Q). “

Furthermore, I find the casual dismissing of leaky expression of the myofiber:iCre#1 line strange. What does that mean? Presumably, that fiber labelling persisted after a 4 dpf Tamoxifen treatment. Yet, without further Tamoxifen there should be no further 'leaky' Cre activity even if the promoter itself in the transgenic is leaky. As I understand it, this has been well demonstrated in both mouse and fish where, even when the Cre is strongly expressed, no recombination occurs until Tamoxifen is added.

Response (2.8): We appreciate the reviewer's comment, but it is important to point out that the CreERT2 leakiness issue is not uncommon, and has been widely acknowledged in both zebrafish and mouse communities (Mosimann and Zon, 2011; Mosimann et al., 2011; Álvarez-Aznar et al., 2020). As even slight leakiness could complicate our cell tracking studies, we thus conducted a years-long screening solely to identify stable transgenic lines devoid of leakiness. Additionally, we have included 'no Tamoxifen' controls for each of our tracking experiments spanning various developmental stages, as detailed on page 12, paragraph 1:

“Of note, we detected zero tagged myofibers in the controls across all the *Tg(palmuscle-Multi; myofiber:iCre#2)* animals examined at different developmental stages (no treatment of Dox and Tam; 14, 29, 42, 43, 70, 180 dpf, 10 mpf, and 16 mpf; Fig. 4J, 5 H-I, 5 L-M, 5P, 6 J-K and SI Appendix, Fig. S3 C-D and S3G; n = 93 animals), a crucial prerequisite for reliable long-term cell fate mapping.”

I do not think that functional and biochemical differences between 14 dpf and 70 dpf fibers argues for replacement. It could just as easily reflect, for example, different electrical activity imposed by the nerve, given the different swim patterns of larvae and adults.

Response (2.9): For clarification, we do not intent to use the findings or differences as an argument for myofiber replacement. Instead, we present these differences as inherent features of the myofibers that exist during key stages of post-embryonic growth (Figure 6).

Dear Chen,

Thank you for submitting your manuscript together with the reviews from another journal and your point-by-point response to them to The EMBO Journal. I have now received input from an arbitrating advisor, who has evaluated the revised manuscript. I have copied the advisor's comments below. As you can see, he/she recommends publication of your study in our journal pending addition of additional discussion on the limitations of the used experimental approach in the manuscript text. Please also consider the advisor's concern regarding acronym use in description of the proposed myofibre replacement process, as I agree that a non-abbreviated terminology is more likely to be used by the subsequent studies.

Additionally, please address the following editorial points before I can extend formal acceptance of the manuscript:

1. Please submit a complete author checklist, which you can download from our author guidelines (<https://www.embopress.org/pb-assets/embo-site/EMBO%20Press%20Author%20Checklist-1642513524327.xlsx>). Please insert information in the checklist that is also reflected in the manuscript. The completed author checklist will also be part of the Review Process File.
2. At EMBO Press we ask authors to provide source data for the main and Expanded View figures. Our source data coordinator will contact you to discuss which figure panels we would need source data for and will also provide you with helpful tips on how to upload and organize the files.
3. Please make sure that the order of the sections in the manuscript is as follows: abstract, introduction, results, discussion, materials & methods, data availability section, acknowledgments, disclosure statement and competing interests, references, main figure legends, tables, expanded figure legends.
4. The highlights and significance statement should be removed from the manuscript.
5. Please turn supplementary figures S1-6 into EV figures. We replaced Supplementary Information with Expanded View (EV) Figures and Tables that are collapsible/expandable online. EV Figures should be cited as 'Figure EV1, Figure EV2' etc. in the text and their respective legends should be included in the main text after the legends of regular figures. Further information on the format is available here: <https://www.embopress.org/page/journal/14602075/authorguide#expandedview>.
6. Please rename the supplemental table into Table EV1.
7. Please rename the movies into Movie EV1-EV7 and update the callouts accordingly. The legends should be removed from the manuscript text file and zipped with each movie file. Further information is available here: <https://www.embopress.org/page/journal/14602075/authorguide#expandedview>
8. CRediT has replaced the traditional author contributions section because it offers a systematic, machine-readable author contributions format that allows for more effective research assessment. Please remove the Author Contributions from the manuscript and use the free text boxes beneath each contributing author's name in our online submission system to add specific details on the author's contribution. More information is available in our guide to authors.
9. Please rename "Declaration of Interests" section into "Disclosure and competing interests statement" (further info: <https://www.embopress.org/page/journal/14602075/authorguide#conflictsofinterest>).
10. Please update references according to The EMBO Journal style - where there are more than 10 authors on a paper, the first 10 should be listed, followed by 'et al.' Please see further information here: <https://www.embopress.org/page/journal/14602075/authorguide#referencesformat>
11. Please update the Data Availability Section according to the journal style. Please remove the "Lead Contact" and "Materials Availability" should be deleted; all source data and primary data generated in this study should be deposited and made freely available. This section should be moved to the end of Methods section. Further information can be found at <https://www.embopress.org/page/journal/14602075/authorguide#dataavailability>
12. Our data editors have flagged the following issues in figure legends that need correcting:
 - Please add information on the number of replicates and their nature in the legends of supplementary figures 2h-i.
 - Please note define the error bars in the legends of supplementary figures 2h-i.
13. Papers published in The EMBO Journal are accompanied online by a 'Synopsis' to enhance discoverability of the manuscript. Please submit a short (1-2 sentences) summary of the findings and their significance in addition to the already provided bullet points highlighting the key results. Please also send us a synopsis image that is 550x300-600 pixels large (width x height, jpeg or png format). You can either show a model or key data in the synopsis image. Please note that the image size is rather small and that text needs to be readable at the final size.

With best wishes,

leva

We realize that it is difficult to revise to a specific deadline. In the interest of protecting the conceptual advance provided by the work, we recommend a revision within 3 months (23rd Jul 2024). Please discuss the revision progress ahead of this time with the editor if you require more time to complete the revisions.

Referee #1:

In this work, Kumar et al., argue that larval myofibers are completely replaced during maturation to adulthood. This is a remarkable observation, which, as Referee 2 highlights merits caution in assessing. The data quality and presentation is high, enabling a thorough examination of the imaging and results.

As this paper has been extensively reviewed previously, I focus on the Reviewer Response and whether the authors have satisfactorily addressed the previous concerns.

The response to Reviewer 1 is clear and not contentious.

Reviewer 2 raises a number of very relevant issues that need to be considered. The claim that (nearly) all larval myofibers are replaced is one that needs careful validation. The authors now provide further evidence that the issue of reporter expression variation is not a problem (Figure 5). They provide three assays to assess fiber loss (imaging, PCR and histological sectioning). While each has limitations, it is reassuring that there is consistency between them. However, they are still relatively coarse - as highlighted, the size later in adults of muscle fibers is a particular barrier to high confidence in the conclusions.

From this, I am inclined to support the authors interpretation, but better caveats need to be made. In particular, I was surprised that there was no paragraph in Discussion "playing Devil's advocate". It would be good to highlight the potential limitations of their approach as this may guide future experiments to validate/argue against these results. Such information should not be confined to the interactions with the reviewers and will also help readers in making their own judgement on the results.

A more minor issue, but I do not like the addition of a new terminology "swap". Biology is already full of acronyms etc., and using such a common word for the process is unhelpful.

Referee #1:

In this work, Kumar et al., argue that larval myofibers are completely replaced during maturation to adulthood. This is a remarkable observation, which, as Referee 2 highlights merits caution in assessing. The data quality and presentation is high, enabling a thorough examination of the imaging and results.

As this paper has been extensively reviewed previously, I focus on the Reviewer Response and whether the authors have satisfactorily addressed the previous concerns.

The response to Reviewer 1 is clear and not contentious.

Reviewer 2 raises a number of very relevant issues that need to be considered. The claim that (nearly) all larval myofibers are replaced is one that needs careful validation. The authors now provide further evidence that the issue of reporter expression variation is not a problem (Figure 5). They provide three assays to assess fiber loss (imaging, PCR and histological sectioning). While each has limitations, it is reassuring that there is consistency between them. However, they are still relatively coarse - as highlighted, the size later in adults of muscle fibers is a particular barrier to high confidence in the conclusions.

From this, I am inclined to support the authors interpretation, but better caveats need to be made. In particular, I was surprised that there was no paragraph in Discussion "playing Devil's advocate". It would be good to highlight the potential limitations of their approach as this may guide future experiments to validate/argue against these results. Such information should not be confined to the interactions with the reviewers and will also help readers in making their own judgement on the results.

(Response 1.1) We thank the reviewer for the input and have included further discussion regarding the limitations of our experimental approach on page 21, paragraph 1:

“Meanwhile, it is important to note that zebrafish can undergo significant growth during the larva-to-adult transition phase, which is expected to reduce the proportion of larval myofibers even in the absence of the replacement. However, we have provided three independent lines of evidence to support the extensive elimination of larval myofibers: 1) continuous live tracking of approximately 500 individual myofibers across two different transgenic lines (Figs. 5A-F and EV3E-I), (2) millimeter-scaled histological examination of the tagged myofibers at single-cell resolution (Figs. 5G-N and EV5D-F), and (3) whole-body qPCR-based detection of the recombined cassettes (Fig. 5R-T’). Altogether, these coherent findings support the occurrence of extensive replacement within the pre-existing larval myofiber populations.”

A more minor issue, but I do not like the addition of a new terminology "swap". Biology is already full of acronyms etc., and using such a common word for the process is unhelpful.

(Response 1.2) Based on the suggestion, we have replaced the acronym SWAP with “whole-body replacement of larval myofibers” throughout the text.

Dear Chen,

Thank you for addressing the final points. I am now pleased to inform you that your manuscript has been accepted for publication. Congratulations on a beautiful study!

Before we forward your manuscript to our publishers, I would like to propose some minor edits in the manuscript abstract and synopsis (please see below and the attached manuscript text file). I have also written a short blurb that will accompany the title of your manuscript in our online system. Please take a look and let me know if any corrections are needed:

Blurb:
Autophagic cell death enables replacement of larval myofibers with their adult counterparts during zebrafish development.

Synopsis:
Post-embryonic development includes myofiber proliferation and growth to supply the needs of the adult organism. Here, extensive tracking of the entire skeletal myofiber population in the zebrafish model reveals that the majority of larval myofibers are replaced by their adult counterparts with distinct morphology, function, and lifespan.

- Development of a myofiber tracking system enables whole-body live monitoring of post-embryonic muscle growth at single-cell resolution.
- Most larval myofibers are eliminated and replaced by adult counterparts during development.
- Larval myofibers and adult myofibers represent two distinct populations.
- Autophagic cell death mediates larval myofiber removal.

If you have any questions, please do not hesitate to contact the Editorial Office. Thank you for this contribution to The EMBO Journal and congratulations on a great paper!

With best wishes,

leva

leva Gailite, PhD
Senior Scientific Editor
The EMBO Journal
Meyerhofstrasse 1
D-69117 Heidelberg
Tel: +4962218891309
i.gailite@embojournal.org
